# MoDGS: Dynamic Gaussian Splatting from Casually-captured Monocular Videos with Depth Priors

**Qingming Liu[1,5]∗**    **Yuan Liu[2]∗**    **Jiepeng Wang[3]**    **Xianqiang Lyv[1]**    **Peng Wang[3]**
**Wenping Wang[4]**    **Junhui Hou[1]†**
[1]City University of HongKong    [2]HKUST    [3]HKU    [4]TAMU    [5]CUHK(SZ)
qingmingliu@foxmail.com    yuanly@ust.hk

## Abstract

In this paper, we propose MoDGS, a new pipeline to render novel views of dynamic scenes from a casually captured monocular video. Previous monocular dynamic NeRF or Gaussian Splatting methods strongly rely on the rapid movement of input cameras to construct multiview consistency but struggle to reconstruct dynamic scenes on casually captured input videos whose cameras are either static or move slowly. To address this challenging task, MoDGS adopts recent single-view depth estimation methods to guide the learning of the dynamic scene. Then, a novel 3D-aware initialization method is proposed to learn a reasonable deformation field and a new robust depth loss is proposed to guide the learning of dynamic scene geometry. Comprehensive experiments demonstrate that MoDGS is able to render high-quality novel view images of dynamic scenes from just a casually captured monocular video, which outperforms state-of-the-art methods by a significant margin. Project page: https://MoDGS.github.io

## 1 Introduction

Novel view synthesis (NVS) is an important task in computer graphics and computer vision, which greatly facilitates downstream applications such as augmented or virtual reality. In recent years, the novel-view-synthesis quality on static scenes has witnessed great improvements thanks to the recent development of techniques such as NeRF (Mildenhall et al., 2020), Instant-NGP (Müller et al., 2022), and Gaussian Splatting (Kerbl et al., 2023), especially when there are sufficient input images. However, novel view synthesis in a dynamic scene with only one monocular video still remains a challenging task.

Dynamic View Synthesis (DVS) has achieved impressive improvements along with the emerging neural representations (Mildenhall et al., 2020) and Gaussian splatting (Kerbl et al., 2023) techniques. Most of the existing DVS methods (Cao & Johnson, 2023; Yang et al., 2023) require multiview videos captured by dense synchronized cameras to achieve good rendering quality. Though some works can process a monocular video for DVS, as pointed out by DyCheck (Gao et al., 2022), these methods require the camera of the monocular video to have extremely large movements, which is called "Teleporting Camera Motion" on different viewpoints, so these methods can utilize the multiview consistency provided by this pseudo multiview video to reconstruct the 3D geometry of the dynamic scene. However, such large camera movements are rarely seen in casually captured videos because casual videos are usually produced by smoothly moving or even static cameras. When the camera moves slowly or is static, the multiview consistency constraint will be much weaker and all these existing DVS methods fail to produce high-quality novel-view images, as shown in Fig. 1.

In this paper, we present Monocular Dynamic Gaussian Splatting (MoDGS) to render novel-view images from casually captured monocular videos in a dynamic scene. MoDGS addresses the weak multiview constraint problem by adopting a monocular depth estimation method (Fu et al., 2024), which provides prior depth information on the input video to help the 3D reconstruction. However, we find that simply applying a single-view depth estimator in DVS to supervise rendered depth

---

∗Equal contribution. This project was primarily completed while Qingming was at CityUHK.

†Corresponding Authors. Email: jh.hou@cityu.edu.hk. This project was supported in part by the NSFC Excellent Young Scientists Fund 62422118, and in part by the Hong Kong Research Grants Council under Grant 11219422 and Grant 11219324.

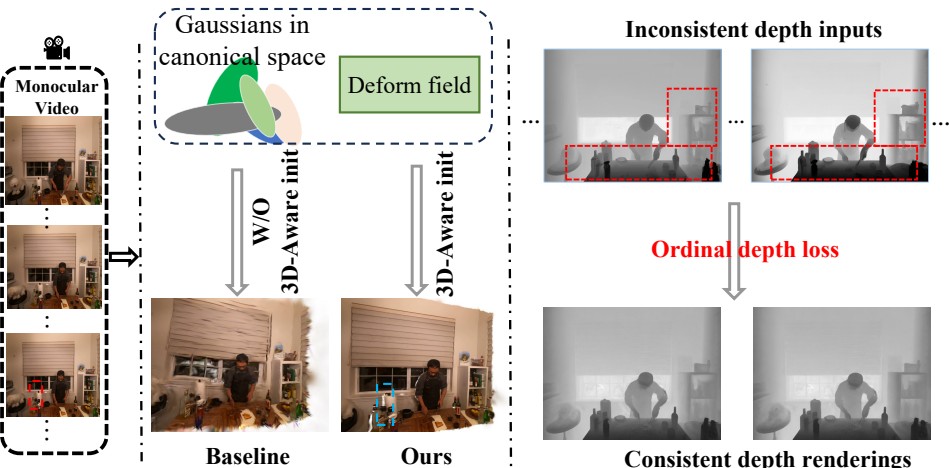

Figure 1: Given a casually captured monocular video of a dynamic scene, **MoDGS** is able to synthesize high-quality novel-view images in this scene. In the middle column, the baseline method (Yang et al., 2023) fails to correctly reconstruct the 3D dynamic scenes on this static monocular video. The white regions in cyan bounding boxes are not visible in the input video (red bounding boxes) so there are some artifacts for these invisible regions. In the rightmost column, the input estimated monocular depth is inconsistent (red bounding boxes); however, our proposed ordinal depth loss effectively ensures more consistent depth outputs. This loss enhances the accuracy and reliability of learning underlying geometry.

maps is not enough for high-quality novel view synthesis. First, the depth supervision only provides information for each frame but does not help to associate 3D points between two frames in time. Thus, we still have difficulty in learning an accurate time-dependent deformation field. Second, the estimated depth values are not consistent among different frames.

To learn a robust deformation field from a monocular video, we propose a 3D-aware initialization scheme for the deformation field. Existing methods (Katsumata et al., 2024) solely rely on supervision from 2D flow estimation, which produces deteriorated results without sufficient multiview consistency. We find that directly initializing the deformation field in the 3D space greatly helps the subsequent learning of the 4D representations and improves the rendering quality as shown in Fig. 1.

To better utilize the estimated depth maps for supervision, we propose a novel depth loss to address the scale inconsistency of estimated depth values across different frames. Previous methods (Li et al., 2023b; Liu et al., 2023a) supervise the rendered depth maps using a scale-invariant depth loss by minimizing the $L2$ distance of normalized rendered depth and depth priors, and the most recent method (Zhu et al., 2023c) proposed to supervise the rendered depth maps using a Pearson correlation loss to mitigate the scale ambiguity between the reconstructed scene and the estimated depth maps. However, the estimated depth maps of different frames are not even consistent after normalizing to the same scale. To address these challenges, we observe that despite the inconsistency in values, the orders of depth values of different pixels in different frames are stable, which motivates us to propose an ordinal depth loss. This novel ordinal depth loss enables us to fully utilize the estimated depth maps for high-quality novel view synthesis.

To demonstrate the effectiveness of MoDGS, we conduct experiments on three widely used datasets, the Nvdia (Yoon et al., 2020) dataset, the DyNeRF (Li et al., 2022) dataset, and the Davis (Pont-Tuset et al., 2017) dataset. We also present results on a self-collected dataset containing monocular in-the-wild videos from the Internet. We adopt an exact monocular DVS evaluation setting that only uses the video of one camera as input while evaluating the video of another camera. Results show that our method outperforms previous DVS methods by a large margin and achieves high-quality NVS on casually captured monocular videos.

## 2 RELATED WORK

In recent years, numerous works have focused on the task of novel view synthesis in both static and dynamic scenes. The main representatives are Neural Radiance Field (Mildenhall et al., 2020) and

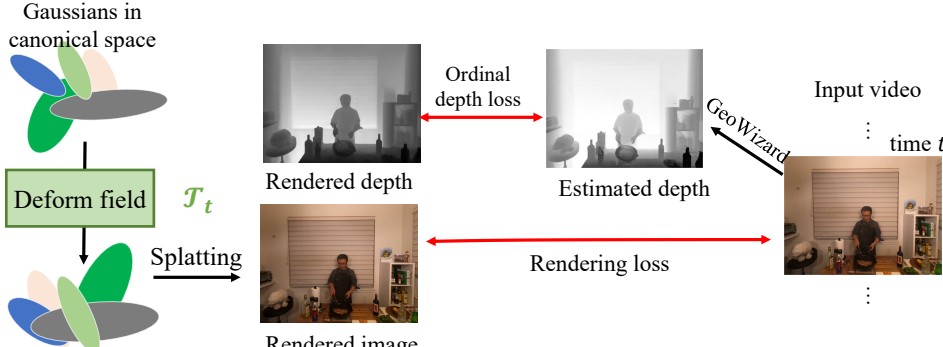

Figure 2: **Overview**. Given a casually captured monocular video of a dynamic scene, MoDGS represents the dynamic scene with a set of Gaussians in a canonical space and a deformation field represented by an MLP $\mathcal{T}$. To render an image at a specific timestamp $t$, we deform all the Gaussians by $\mathcal{T}_t$ and then use the splatting technique to render images and depth maps. While in training MoDGS, we use a single-view depth estimator GeoWizard (Fu et al., 2024) to estimate depth maps and compute the rendering loss and an ordinal depth loss for training.

Gaussian Splatting (Kerbl et al., 2023), along with their variants. In this paper, we primarily focus on view synthesis in dynamic scenes.

**Dynamic NeRF.** Recent dynamic NeRF methods can be roughly categorized into two groups. 1) Representing by time-varying neural radiance fields conditioned on time (Gao et al., 2021; Li et al., 2022; Park et al., 2023). For example, Park et al. (2023) proposes a simple spatiotemporal radiance field by interpolating the feature vectors indexed by time. 2) Representing by a canonical space NeRF and deformation field (Guo et al., 2023; Li et al., 2021; Park et al., 2021a;b; Pumarola et al., 2021; Tretschk et al., 2021; Xian et al., 2021). For example, NSFF (Li et al., 2021) models the dynamic components using forward and backward flow represented as 3D dense vector fields; Nerfies (Park et al., 2021a) and HyperNeRF (Park et al., 2021b) model the scene dynamics as a deformation field mapping to a canonical space. Recent advances in grid-based NeRFs (Müller et al., 2022; Sara Fridovich-Keil and Alex Yu et al., 2022; Chen et al., 2022) demonstrate that the training of static NeRFs can be significantly accelerated. Consequently, some dynamic NeRF works utilize these grid-based or hybrid representations for fast optimization (Guo et al., 2023; Cao & Johnson, 2023; Fang et al., 2022; Fridovich-Keil et al., 2023; Shao et al., 2023; Wang et al., 2023a;b; Song et al., 2023; You & Hou, 2023).

**Dynamic Gaussian Splatting.** The recent emergence of 3D Gaussian Splatting (3DGS) demonstrates its efficacy for super-fast real-time rendering attributed to its explicit point cloud representation. Recent follow-ups extend 3DGS to model dynamic 3D scenes. Luiten et al. (2023) track dynamic 3D Gaussians by frame-by-frame training from synchronized multi-view videos. Yang et al. (2023) propose a deformable version of 3DGS by introducing a deformation MLP network to model the 3D flows. Wu et al. (2024) and Duisterhof et al. (2023) also introduce a deformation field but using a more efficient Hexplane representation (Cao & Johnson, 2023). Yang et al. (2024c) proposes a dynamic representation with a collection of 4D Gaussian primitives, where the time evolution can be encoded by 4D spherical harmonics. Bae et al. (2024) encodes motions with a per-Gaussian feature vector. Some other works (Li et al., 2023a; Lin et al., 2024; Liang et al., 2023) also study how to effectively encode the motions for Gaussians with different bases. To effectively learn the motions of Gaussians, some works (Feng et al., 2024; Yu et al., 2023; Huang et al., 2024) resort to clustering the motions together for a compact representation.

**DVS from Casual Monocular Videos.** As shown in DyCheck (Gao et al., 2022), many existing monocular dynamic view synthesis datasets used for benchmarking, like D-NeRF (Pumarola et al., 2021), HyperNeRF (Park et al., 2021b), and Nerfies (Park et al., 2021a), typically involve significant camera movements between frames but with small object dynamic motions. While this capture style helps with multi-view constraints and dynamic 3D modeling, it is not representative of casual everyday video captures. When using casual videos, the reconstruction results from these methods suffer from quality degradation. Some works address dynamic 3D scene modeling using monocular casual videos. DynIBaR (Li et al., 2023b) allows for long-sequence image-based rendering of

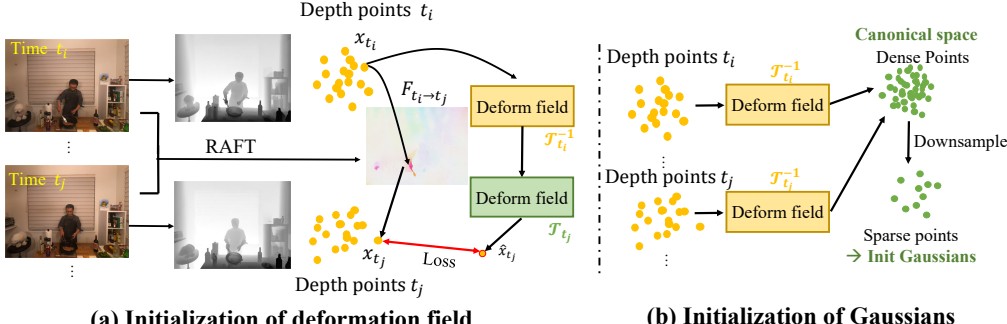

**(a) Initialization of deformation field**    **(b) Initialization of Gaussians**

Figure 3: (a) **Initialization of the deformation field**. We first lift the depth maps and a 2D flow to a 3D flow and train the deformation field for initialization. (b) **Initialization of Gaussians in the canonical space**. We use the initialized deformation field to deform all the depth points to the canonical space and downsample these depth points to initialize Gaussians.

dynamic scenes by aggregating features from nearby views, but its training cost is high for long per-scene optimization. Lee et al. (2023) proposes a hybrid representation that combines static and dynamic elements, allowing for faster training and rendering, though it requires additional per-frame masks for dynamic components. RoDynRF (Liu et al., 2023a) focuses on robust dynamic NeRF reconstruction by estimating NeRF and camera parameters together. DpDy (Wang et al., 2024a) enhances quality by fine-tuning a diffusion model with SDS loss supervision (Poole et al., 2022), but it demands significant computational resources. Concurrent works like DG-Marbles (Stearns et al., 2024) use Gaussian marbles and a hierarchical learning strategy to optimize representations. Shape-of-Motion (Wang et al., 2024b) and Mosca (Lei et al., 2024) rely on explicit motion representation and initialize scene deformation with depth estimation and video tracking priors. However, our MoDGS method effectively uses only noisy inter-frame flow maps from RAFT (Teed & Deng, 2020) as input, performing well without the need for strong long-range pixel correspondence.

**Ordinal Relation in Depth Maps.** The ordinal relation between pixels has been investigated in recent years, especially in the field of monocular depth estimation. Zoran et al. (2015) proposed to use a three-category classification network to predict the order relation of given pixel pairs. Then the depth can be extracted by optimizing a constrained quadratic optimization problem. Similarly, Fu et al. (2018) treats the depth prediction problem as a multi-class classification problem. Chen et al. (2016) further proposes a ranking loss to learn metric depth, which encourages a small difference between depths if the ground-truth relation is equal; otherwise it encourages a large difference. Then, Pavlakos et al. (2018) extends this differentiable ranking loss to the human pose estimation task. However, these works only utilize limited numbers of depth orders for training (one pair in Chen et al. (2016) and 17 pairs in Pavlakos et al. (2018)), resulting in coarse supervision for depth maps. The direct application of their ranking loss as depth supervision has yet to be explored. Moreover, our ordinal depth takes the rendered metric depth maps as input, which are dense grids of float numbers. Our task is different from previous depth estimation and pose estimation tasks and we present a comparison between our ordinal depth loss and depth ranking loss in Appendix A.10.

## 3    PROPOSED METHOD

Given a casually captured monocular video, we aim to synthesize novel view images from this video. We propose MoDGS, which achieves this by learning a set of Gaussians $\{G_i | i = 1, \cdots, N\}$ in a canonical space and a deformation field $\mathcal{T}_t : \mathbb{R}^3 \to \mathbb{R}^3$ to deform these Gaussians to a specific timestamp $t$. Then, for a timestamp $t$ and a camera pose, we use splatting to render an image.

**Overview.** As shown in Fig. 2, to train MoDGS, we split the monocular video into a sequence of images $\{I_t | t = 1, ..., T\}$ with known camera poses. We denote our deformation field as a function $x_t = \mathcal{T}_t(x)$, which maps a 3D location $x \in \mathbb{R}^3$ in the canonical 3D space to a location $x_t \in \mathbb{R}^3$ in the 3D space on time $t$. For every image $I_t$, we utilize a single-view depth estimator (Fu et al., 2024) to estimate a depth map $D_t$ for every image and utilize a flow estimation method RAFT (Teed & Deng, 2020) to estimate a 2D optical flow $F_{t_i \to t_j}$ between $I_{t_i}$ and $I_{t_j}$ where $t_i$ and $t_j$ are two arbitrary timestamps. Then, we initialize our deformation field by a 3D-aware initialization scheme as introduced in Sec. 3.2. After initialization, we train our Gaussians and deformation field with a

rendering loss and a new depth loss introduced in Sec. 3.3. In the following, we first begin with the definition of the Gaussians and the rendering process in MoDGS.

### 3.1 GAUSSIANS AND DEFORMATION FIELDS

**Gaussians in the canonical space.** We define a set of Gaussians in the canonical space, we follow the original 3D GS (Kerbl et al., 2023) to define a 3D location, a scale vector, a rotation, and a color with spherical harmonics. Note this canonical space does not explictly correspond to any timestamp but is just a virtual space that contains the canonical locations of all Gaussians.

**Deformation fields.** The deformation field $\mathcal{T}_t$ used in MoDGS follows the design of Omnimotion (Wang et al., 2023c) and CaDeX (Lei & Daniilidis, 2022) which is an invertible MLP network (Dinh et al., 2016). This is an invertible MLP means that both $\mathcal{T}_t$ and $\mathcal{T}_t^{-1}$ can be directly computed from the MLP network. All $\mathcal{T}_t$ at different timestamps $t$ share the same MLP network and the time $t$ is normalized to $[0, 1]$ as input to the MLP network.

**Render with MoDGS.** After training both the Gaussians in canonical space and the deformation field, we will use the deformation field to deform the Gaussians in the canonical space to a specific time step $t$. Then, we follow exactly the splatting techniques in 3D GS (Kerbl et al., 2023) to render images from arbitrary viewpoints.

### 3.2 3D-AWARE INITIALIZATION

Original 3D Gaussian splatting (Kerbl et al., 2023) relies on the sparse points from Structure-from-Motion (SfM) to initialize all the locations of Gaussians. When we only have a casually captured monocular video, it is difficult to get an initial set of sparse points for initialization from SfM. Though it is possible to initialize all the Gaussians from the points of the estimated single-view depth of the first frame, we show that this leads to suboptimal results. At the same time, we need to initialize not only the Gaussians but also the deformation field. Thus, we propose a 3D-aware initialization scheme for MoDGS.

**Initialization of depth scales.** Since the estimated depth maps on different timestamps would have different scales, we first estimate a coarse scale for every frame to unify the scales. We achieve this by first segmenting out the static regions on the video and then computing the scale with a least square fitting (Chung et al., 2023b). The static regions can be determined by either thresholding on the 2D flow (Teed & Deng, 2020) or segmenting with a segmentation method like SAM2 (Ravi et al., 2024). Then, on these static regions, we reproject the depth values at a specific timestamp to the first frame and minimize the difference between the projected depth and the depth of the first frame, which enables us to solve for a scale for every frame. We rectify all depth maps with the computed scales. In the following, we reuse $D_t$ to denote the rectified depth maps by default.

**Initialization of the deformation field.** As shown in Fig. 3 (left), given two depth maps $D_{t_i}$ and $D_{t_j}$ along with the 2D flow $F_{t_i \to t_j}$, we lift them to a 3D flow $F_{t_i \to t_j}^{3D}$. This is achieved by first converting the depth maps into 3D points in the 3D space. Then, the estimated 2D flow $F_{t_i \to t_j}$ actually associate two sets of 3D points, which results in a 3D flow $F_{t_i \to t_j}^{3D}$. After getting this 3D flow, we then train our deformation field $\mathcal{T}$ with this 3D flow. Specifically, for a pixel in $I_{t_i}$ whose corresponding 3D point is $x_{t_i}$, we query $F_{t_i \to t_j}^{3D}$ to find its target point $x_{t_j}$ in the $t_j$ timestamp. Then, we minimize the difference by

$$\ell_{\text{init}} = \sum \|\mathcal{T}_{t_j} \circ \mathcal{T}_{t_i}^{-1}(x_{t_i}) - x_{t_j}\|^2. \tag{1}$$

We train the MLP in $\mathcal{T}$ for a fixed number of steps to initialize the deformation field.

**Initialization of Gaussians.** After getting the initialized deformation field, we will initialize a set of 3D Gaussians in the canonical space as shown in Fig. 3 (right). We achieve this by first converting all the depth maps to get 3D points. Then, these 3D points are deformed backward to the canonical 3D space. This means that we transform all the depth points of all timestamps to the canonical space, which results in a large amount of points. We then evenly downsample these points with a predefined voxel size to reduce the point number and we initialize all our Gaussians with the locations of these downsampled 3D points in the canonical space. Here, more advanced learning-based adaptive fusion strategies (You et al., 2023) could be adopted to downsample the points, potentially improving the representation.

### 3.3 ORDINAL DEPTH LOSS

**Pearson correlation loss.** Existing dynamic Gaussian Splatting or NeRF methods also adopt a depth loss to supervise the learning of their 3D representations. One possible solution (Zhu et al., 2023c; Li et al., 2021; Liu et al., 2023a) is to maximize a Pearson correlation between the rendered depth and the estimated single-view depth

$$\text{Corr}\left(\hat{D}_t, D_t\right) = \frac{\text{Cov}\left(\hat{D}_t, D_t\right)}{\sqrt{\text{Var}\left(\hat{D}_t\right)\text{Var}(D_t)}},$$

where $\text{Cov}\left(\cdot, \cdot\right)$ and $\text{Var}\left(\cdot, \cdot\right)$ means the covariance and variance respectively, $D_t$ and $\hat{D}_t$ are the estimated depth and the rendered depth respectively. Since the estimated single-view depth has ambiguity in scale, the Pearson correlation loss avoids the negative effects of the scale ambiguity. Note that in Li et al. (2021) and Liu et al. (2023a), the loss is called normalized depth loss, which is equivalent to Pearson correlation here as shown in the supplementary material.

**Limitations of Pearson correlation loss.** However, we find that this Pearson correlation depth loss is still suboptimal. As shown in Fig. 4, the estimated depth maps at two different timestamps are still not consistent with each other after normalization. Making two depth maps consistent after normalization actually requires these two depth maps to be related by a linear transformation, i.e. $D_{t+1} = aD_t + b$ with $a$ and $b$ two constants. However, the single-view depth estimation method is not accurate enough to guarantee the linear relationship between two estimated depth maps at different timestamps. In this case, the Pearson correlation loss still brings inconsistent supervision.

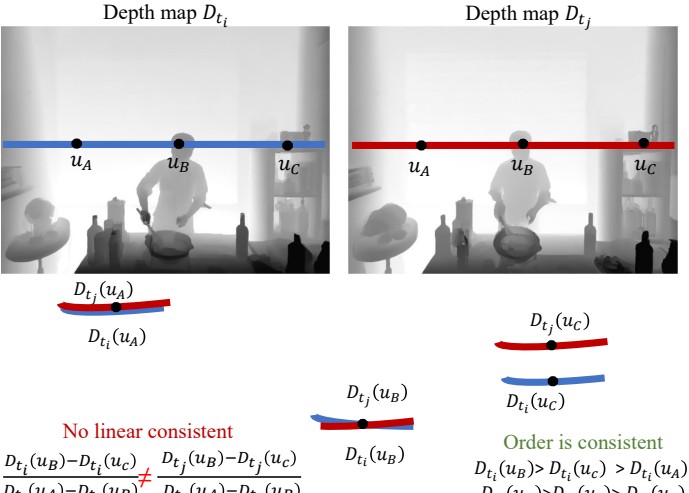

Figure 4: We show the estimated single-view depth maps at two different timestamps $D_{t_i}$ and $D_{t_j}$ after normalization to the same scale. Since the single-view depth estimator is not accurate enough, the depth maps are not linear related so the scale normalization does not perfectly align them. However, the order of depth values on three corresponding pixels is stable for these two depth maps, which motivates us to propose an ordinal depth loss for supervision.

**Ordinal depth loss.** To address this problem, our observation is that though we cannot guarantee depth consistency after normalization, as shown in Fig. 4, the order of depth value is consistent among two different frames. Thus, this motivates us to ensure the order of depth is correct by a new ordinal depth loss. We first define an order indicator function

$$\mathcal{R}(D_t(u_1), D_t(u_2)) = \begin{cases} +1, & D_t(u_1) > D_t(u_2) \\ -1, & D_t(u_1) < D_t(u_2) \end{cases}, \tag{2}$$

where $\mathcal{R}$ is the order indicator function on depth map $D_t$ which indicates the order between the depth values of pixels $u_1 \in \mathbb{R}^2$ and $u_2 \in \mathbb{R}^2$, and $D_t(u)$ means the depth value on the pixel $u$. Then, we define our ordinal depth loss based on the depth order by

$$\ell_{\text{ordinal}} = \|\tanh\left(\alpha(\hat{D}_t(u_1) - \hat{D}_t(u_2))\right) - \mathcal{R}\left(D_t(u_1), D_t(u_2)\right)\|, \tag{3}$$

where $\tanh(x) = \frac{e^x - e^{-x}}{e^x + e^{-x}}$, $\hat{D}_t$ means the rendered depth map at timestamp $t$, $\hat{D}_t(u)$ is the depth value of this rendered depth map on the pixel $u$, $\alpha$ is a predefined constant. Eq. 3 means we transform the depth difference between $\hat{D}_t(u_1)$ and $\hat{D}_t(u_2)$ to 1 or -1 by tanh function. Then, we force the depth order of the rendered depth map $\hat{D}_t$ to be consistent with the order in the predicted depth map $D_t$. In the implementation, we randomly sample 100k pairs $(u_1, u_2)$ to compute the ordinal depth loss.

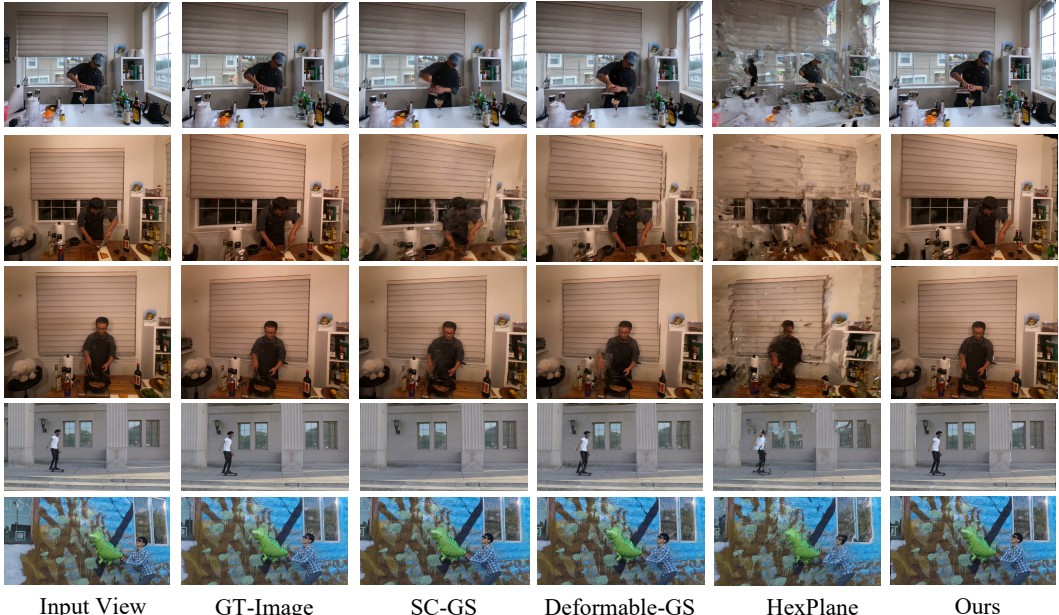

| Input View | GT-Image | SC-GS | Deformable-GS | HexPlane | Ours |

Figure 5: Qualitative comparison on the novel-view renderings of the DyNeRF (Li et al., 2022) and Nvidia (Yoon et al., 2020) datasets. We compare MoDGS with SC-GS (Huang et al., 2024), Deformable-GS (Yang et al., 2023), and HexPlane (Cao & Johnson, 2023).

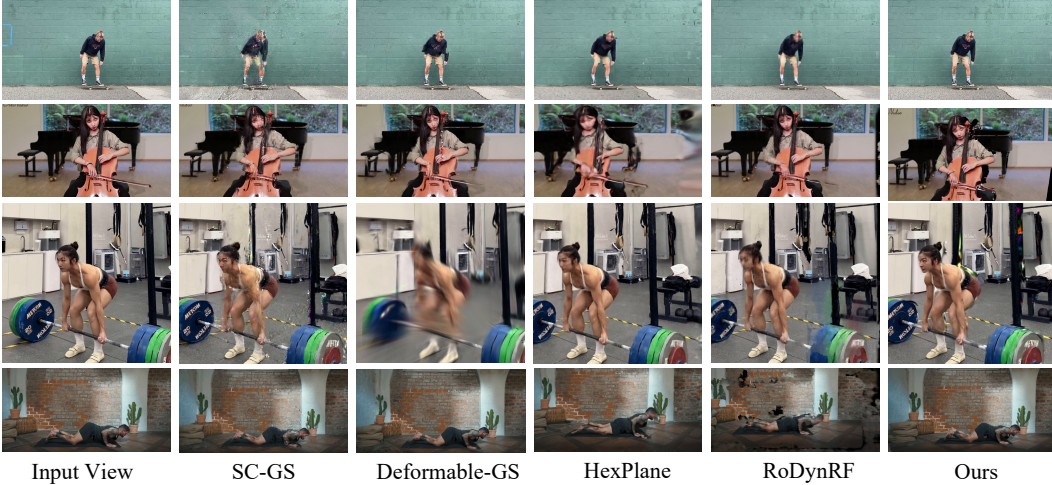

| Input View | SC-GS | Deformable-GS | HexPlane | RoDynRF | Ours |

Figure 6: Qualitative comparison of DVS quality on the MCV dataset. We compare MoDGS with SC-GS (Huang et al., 2024), Deformable-GS (Yang et al., 2023), HexPlane (Cao & Johnson, 2023), and RoDynRF (Liu et al., 2023a).

### 3.4 TRAINING OF MODGS

After initializing the Gaussians and the deformation fields, we use MoDGS to render at a specific timestamp and compute the rendering loss $\ell_{render}$ and the ordinal depth loss $\ell_{ordinal}$. So the total training loss for MoDGS is

$$\ell = \lambda_{ordinal}\ell_{ordinal} + \lambda_{render}\ell_{render}. \tag{4}$$

## 4 EXPERIMENTS

### 4.1 EVALUATION PROTOCOLS

**Datasets.** We conducted experiments on four datasets to demonstrate the effectiveness of our method. The first dataset is the DyNeRF (Li et al., 2022) dataset which consists of 6 scenes. On each scene, we have 18-20 synchronized cameras capturing 10-30 second videos. In these videos, there is

Table 1: Quantitative results on the DyNeRF (Li et al., 2022) and Nvidia (Yoon et al., 2020) datasets. We compare our method with SC-GS (Huang et al., 2024), Deformable GS (Yang et al., 2023) (D-GS) and HexPlane (Cao & Johnson, 2023) in PSNR↑, SSIM↑, and LPIPS↓. For a per-scene breakdown of the metrics, please refer to Table 6 and Table 7 in A.9.

| Methods | DyNeRF | | | Nvidia | | |
| | PSNR↑ | SSIM↓ | LPIPS↓ | PSNR↑ | SSIM↓ | LPIPS↓ |
|---|---|---|---|---|---|---|
| HexPlane | 15.33 | 0.5593 | 0.4514 | 17.17 | 0.3675 | 0.4756 |
| SC-GS | 18.77 | 0.7359 | 0.2310 | 17.59 | 0.4679 | 0.3348 |
| D-GS | 19.55 | 0.7446 | 0.2171 | 18.07 | 0.4650 | 0.3422 |
| Ours | **22.64** | **0.8042** | **0.1545** | **19.27** | **0.5235** | **0.2581** |

mainly a man working on a desktop, like cutting beef or dumping water. We use camera0 for training and evaluate the results on camera5 and camera6. The second dataset is the Nvidia (Yoon et al., 2020) dataset which contains more diverse dynamic subjects like jumping, playing with balloons, and opening an umbrella. The Nvidia dataset contains 8 scenes, which also has 12 synchronized cameras. We train all methods on camera4 and evaluate with camera3 and camera5. Besides, we also collect 6 online videos to construct an in-the-wild dataset, called the Monocular Casual Video (MCV) Dataset, to demonstrate our method can generalize to in-the-wild casual videos. The MCV dataset contains diverse subjects like skating, a dog eating food, YOGA, etc. The MCV dataset only contains a single video for each scene, so we cannot evaluate the quantitative results but only report the qualitative results on this dataset. We also present results of the Davis dataset (Pont-Tuset et al., 2017) in Sec. A.6 of the appendix.

**Evaluation setting.** Previous DVS methods (Cao & Johnson, 2023; Yang et al., 2023; Gao et al., 2021) all use different cameras to train the dynamic NeRF or Gaussian Splatting. Even though they only use one camera at a specific timestamp, they use different cameras at different timestamps so that a pseudo multiview video can be constructed to learn the 3D structures of the scene. Since our target is to conduct novel-view synthesis on the casually captured images, we do not adopt such "teleporting camera motions" to construct training videos but just adopt one static camera to record training videos. Then, we render the images from the viewpoints of another camera for evaluation.

**Metrics.** To evaluate the rendering quality, we have to render images from a new viewpoint and compare them with the ground-truth images. However, if the input video is almost static, the input video will contain insufficient 3D information and there would exist an ambiguity in scale. Thus, different DVS methods would choose different scales in reconstructing dynamic scenes so that the rendered images on the novel viewpoints are not aligned with the given ground-truth images. To address this problem, we manually label correspondences between the training images and ground-truth novel-view images. Then, we render a depth map on the training image using the reconstructed dynamic scene and optimize for a scale factor to scale the depth value to satisfy these labeled correspondences. After aligning the scale factors of different methods with the ground-truth images, we compute the SSIM, LPIPS, and PSNR between the rendered images and the ground-truth images.

**Baseline methods.** We compare MoDGS with 4 baseline methods to demonstrate the superior ability of MoDGS to synthesize novel-view images with casually captured monocular videos. These methods can be categorized into two classes. The first is the NeRF-based methods including Hex-Plane (Cao & Johnson, 2023) and RoDynRF (Liu et al., 2023a). HexPlane represents the scene with six feature planes in both 3D space and time-space. We find that HexPlane does not produce reasonable results if only a monocular video from a single camera is given as input. Thus, other than the input monocular video, we use another video from a different viewpoint to train HexPlane for the DVS task. RoDynRF is a SoTA NeRF-based DVS method that also adopts single-view depth estimation as supervision for the 3D dynamic representations. We train it with the same single-view depth estimator GeoWizard (Fu et al., 2024) as ours. The second class is the Gaussian Splatting-based DVS methods including Deformable-GS (Yang et al., 2023) and SC-GS (Huang et al., 2024). Deformable-GS also associates a deformation field with a set of canonical Gaussians for DVS. SC-GS learns a set of keypoints and uses the deformation of these keypoints to blend the deformation of arbitrary 3D points.

### 4.2 COMPARISON WITH BASELINES

The qualitative results on the DyNeRF and Nvidia datasets are shown in Fig. 5. Other qualitative results on our MCV dataset are shown in Fig. 6. The quantitative results on the Nvidia and DyNeRF datasets are shown in Table 1. **Supplementary videos contain more comparison results.**

Synthesizing novel views from a casually captured monocular video is a challenging task. As shown in Fig. 5, though baseline methods achieve impressive results on these benchmarks with "teleporting camera motions", these methods fail to correctly reconstruct the 3D geometry of the dynamic scenes and produce obvious artifacts on both dynamic foreground and static background. The main reason is that the monocular camera is almost static and does not provide enough multiview consistency to reconstruct high-quality 3D geometry for novel view synthesis. In the third row of Fig. 5, SC-GS (Huang et al., 2024) fails to reconstruct the dynamic foreground subject because SC-GS has an initialization process that treats the whole scene as a static scene and trains on the scene for a number of steps. When the foreground subject is moving with a large motion (like skating from left to right), it would be ignored by the static scene initialization and then we fail to reconstruct in the subsequent steps.

In comparison, our method relies on a 3D-aware initialization which provides a strong basis for the subsequent optimization. Meanwhile, our ordinal depth loss enables the 3D prior from the single-view depth estimator for an accurate reconstruction of the dynamic scenes. The quantitative results in Table 1 also show that our method achieves the best performances in all metrics on both datasets. Note that there are still some artifacts on occlusion boundaries because the input monocular camera is almost static and these regions are not visible in our input videos.

### 4.3 ABLATION STUDIES

We conduct ablation studies with our initialization and depth loss on the DyNeRF (Li et al., 2022) dataset to demonstrate their effectiveness. The qualitative results are shown in Fig. 7 and Fig. 8 while the quantitative results are shown in Table 2. In the appendix, we also provide a comparison with depth warping, robustness to depth noises, a discussion on video depth, a comparison with other depth ranking losses, and a discussion about alpha values.

#### 4.3.1 3D-AWARE INITIALIZATION

To show the effectiveness of our 3D-aware initialization, we adopt a random initialization for the deformation field. Based on the random initialization, we still deform all the depth points backward to the canonical space and downsample these points to initialize the Gaussians. Then, we follow the exact same training procedure to train the randomly initialized baseline method. The final results of this random

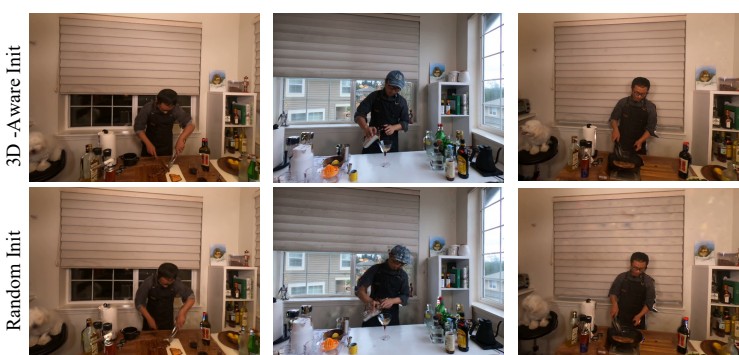

Figure 7: Visualization of rendering results using and without using our 3D-aware initialization.

initialization are shown in Fig. 7. In comparison with our 3D-aware initialization, this random initialization produces more artifacts on the dynamic foreground human, which demonstrates that our 3D-aware initialization provides a good initial point for subsequent 3D dynamic scene reconstruction.

#### 4.3.2 ORDINAL DEPTH LOSS

To demonstrate the effectiveness of our proposed ordinal depth, we experiment with two baseline settings, removing the depth loss and utilizing the Pearson correlation as the depth loss. As shown in Fig. 8, when there is no depth loss, the reconstructed 3D geometry contains much noise and obvious artifacts exist in the scene. Since we perform the 3D-aware initialization on this experiment to ensure a fair comparison, the results without depth loss still show reasonable rendered depth maps. Adopting the Pearson depth loss improves the quality by linear correlating the rendered depth maps

and input depth maps but still produces noisy depth inside a region. In comparison, our ordinal depth loss enables a smooth reconstruction of the depth map in the interior while maintaining sharp edges at boundaries. Thus, the proposed ordinal depth loss enables a more robust reconstruction of the dynamic scene.

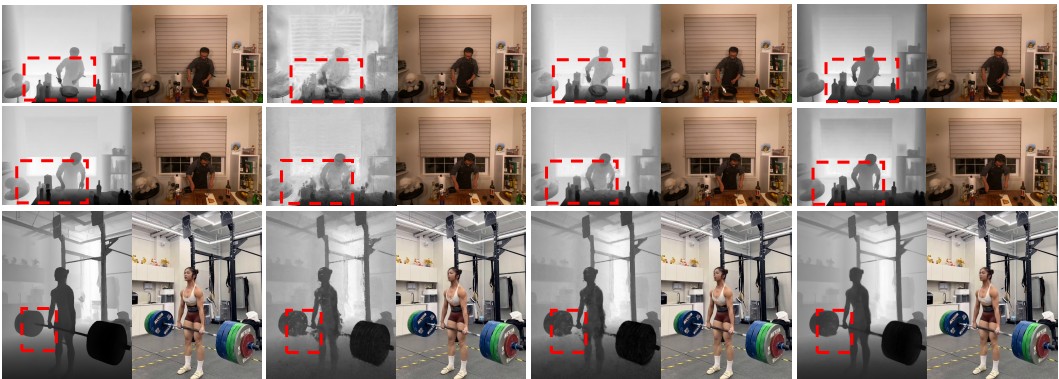

Input Truth      No Depth Loss      Pearson Correlation Loss      Ordinal Depth Loss

Figure 8: Visualization of the rendered depth and RGB images using our ordinal depth loss and the Pearson correlation loss.

## 4.4 LIMITATIONS

Though our method can conduct dynamic view synthesis from casually captured monocular videos, the task is still extremely challenging. One limitation is that our method can only reconstruct the visible 3D parts but cannot imagine the unseen parts, which leads to artifacts when rendering novel view videos on these unseen parts. Incorporating recent 3D-related diffusion generative models (Chung et al., 2023a; Liu et al., 2023b; Long et al., 2023) could be a promising direction to solve this problem, which we leave for future works. Another limitation is that the current training time is comparable with existing DVS methods, which could take several hours for a single scene. How to efficiently reconstruct the dynamic field would be an interesting and promising future research topic. Meanwhile, when the camera is completely static, our method strongly relies on the single-view depth estimator to estimate the 3D depth maps. Though existing single-view depth estimators (Fu et al., 2024; Ke et al., 2023; Yang et al., 2024a; Bhat et al., 2023) are trained on large-scale datasets and predict reasonable depth maps for most cases, these depth estimators may fail to capture some details which degenerate the quality. Another limitation is that we assume consistent depth order, though this assumption is satisfied by most methods (Yang et al., 2024b; Bochkovskii et al., 2024; Ke et al., 2023; Fu et al., 2024). Additionally, a challenge arises in rapidly moving scenes. Such rapid motions may result in inaccurate camera pose estimation, which is still challenging for MoDGS and all existing dynamic Gaussian reconstruction methods. Meanwhile, scenes featuring heavy specular reflections and low-light conditions cannot be well handled, which is also an extremely challenging problem in the monocular setting, which we leave for future works.

Table 2: Ablation studies with the 3D-aware initialization ("3D-aware Init") and Depth Loss on the DyNeRF (Li et al., 2022) dataset. "Ordinal" means the ordinal depth loss while "Pearson" means the Pearson correlation loss.

| 3D-aware Init | Loss | PSNR↑ | SSIM↑ | LPIPS↓ |
|---|---|---|---|---|
| × | Ordinal | 21.27 | 0.7655 | 0.1984 |
| ✓ | Pearson | 21.77 | 0.7938 | 0.1680 |
| ✓ | Ordinal | **22.96** | **0.8103** | **0.1518** |

## 5 CONCLUSION

In this paper, we have presented a novel dynamic view synthesis paradigm called MoDGS. In comparison with existing DVS methods which requires "teleporting camera motions", MoDGS is designed to render novel view images from a casually captured monocular video. MoDGS introduces two new designs to finish this challenging task. First, a new 3D-aware initialization scheme is proposed, which directly initializes the deformation field to provide a reasonable starting point for subsequent optimization. We further analyze the problem of depth loss and propose a new ordinal depth loss to supervise the learning of the scene geometry. Extensive experiments on three datasets demonstrate superior performances of our method on in-the-wild monocular videos over baseline methods.

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

## A  APPENDIX

### A.1  IMPLEMENTATION DETAILS

We implement our MoDGS with PyTorch. To initialize the deformation field, we train it with 20k steps as stated in Sec. 3.2. Subsequently, we jointly train the 3D Gaussians and the deformation field with the rendering loss and the ordinal depth loss for another 20k steps. In Sec. 3.2, the flow is computed in evenly sampled key frames(e.g., $1/5$). And the downsampling voxel size for Gaussian initialization is $0.004^3$ (scenes are normalized to $[-1, 1]^3$ ). For the outer optimization loop and rendering loss, we exactly follow the original 3DGS. And we use Gaussian centers to render depth (Yang et al., 2023). We adopt an Adam optimizer for optimization. The learning rate for 3D Gaussians exactly follows the official implementation of 3D GS (Kerbl et al., 2023), while the learning rate of the deformation network undergoes exponential decay from 1e-3 to 1e-4 in initialization and from 1e-4 to 1e-6 in the subsequent optimization. We set $\alpha = 100$ for $\ell_{\text{ordinal}}$. The weight of our depth order loss is 0.1. When computing depth ordinal loss, we first normalize the depth range to [0,1] and we only consider the depth pair with a difference larger than 0.02 for loss computation. The whole training takes around 3.5 hours to converge (2 hours for the initialization and 1.5 hours for the subsequent optimization) on an NVIDIA RTX A6000 GPU, which uses about 14G memory. The rendering speed of MoDGS is about 75 FPS.

### A.2  REAL DEPTH RECOVERY

The depth prediction of GeoWizard (Fu et al., 2024) is normalized depth values in [0,1] and we have to transform them into real depth values. We follow their official implementation to estimate a scaling factor and an offset value on the normalized normal maps by minimizing the normal maps estimated from the transformed real depth values and the estimated normal maps from GeoWizard. The optimization process takes just several seconds. Note that even after this normalization, the depth maps of different timestamps still differ from each other in scale.

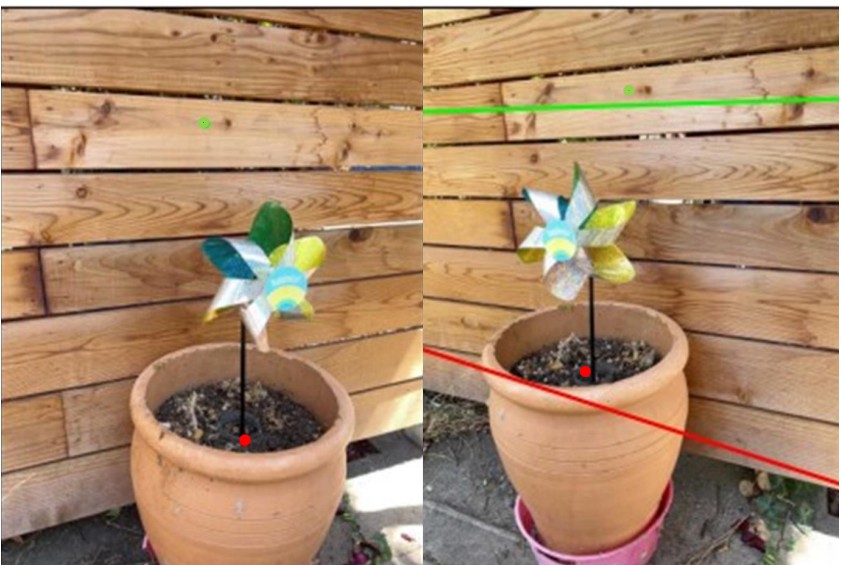

Figure 9: We provide a pair of images on the iPhone (Gao et al., 2022) dataset. We draw two correspondences in the same color and their corresponding epipolar lines that are computed from the provided camera poses. The epipolar lines deviate far from the correspondences, which demonstrate that the camera poses are not accurate enough.

### A.3  POSE ERRORS IN THE IPHONE DATASET

DyCheck (Gao et al., 2022) adopts the iPhone dataset as the evaluation dataset for the DVS on casually captured videos. We do not adopt this dataset because we find that the poses on this dataset

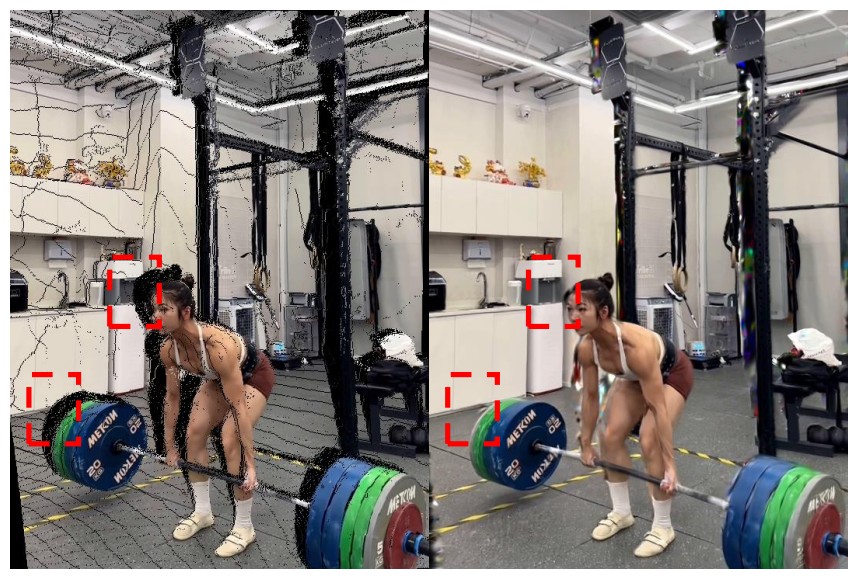

Depth Warping                    Ours

Figure 10: Comparison of our renderings and depth warping. Our method accumulates information among different timestamps and thus is able to render more completed images.

are not very accurate. An example is shown in Fig. 9, where we draw two correspondences and their corresponding epipolar line in the same color. The epipolar line is computed from the provided camera poses. As we can see, the epipolar line does not pass through the correspondence, which means that the provided poses are not accurate enough. We have tried to rerun COLMAP on the iPhone dataset but cannot get reasonable results.

## A.4 DIFFERENCE FROM DEPTH MAP WARPING

MoDGS learns a set of 3D Gaussians in a canonical space and a deformation field to transform it to an arbitrary timestamp. This means that MoDGS is able to accumulate information among different timestamps to reconstruct a more completed scene than just using a single-view depth estimation. We show the difference between our renderings and just warping the training view using the estimated single-view depth map in Fig. 10. As we can see, MoDGS produces more completed reconstruction on contents that are not visible on this timestamp. Meanwhile, MoDGS rectifies the single-view depth maps to be more accurate so the rendering quality is much better.

## A.5 PEARSON DEPTH LOSS AND SCALE-SHIFT INVARIANT DEPTH LOSS

In this section, we will prove that the Pearson Depth loss is equivalent to Scale-Shift Invariant loss. Previous works (Li et al., 2021; Liu et al., 2023a) use scale and shift invariant depth loss by minimizing the $L2$ distance of normalized ground truth depth map $\text{Norm}(D_t)$ and the normalized rendered depth map $\text{Norm}(\hat{D}_t)$

$$\ell_{\text{depth}} = \|\text{Norm}(D_t) - \text{Norm}(\hat{D}_t)\| \tag{5}$$

where $\text{Norm}(D_t)$ denotes $\dfrac{D_t - u_{D_t}}{\sigma_{D_t}}$.

$$\left( \frac{D_t - u_{D_t}}{\sigma_{D_t}} - \frac{D_t - u_{\hat{D}_t}}{\sigma_{\hat{D}_t}} \right)^2$$

$$= -\frac{2(D_t - u_{D_t})(D_t - u_{\hat{D}_t})}{\sigma_{D_t}\sigma_{\hat{D}_t}} + \left( \frac{D_t - u_{D_t}}{\sigma_{D_t}} \right)^2 + \left( \frac{\hat{D}_t - u_{\hat{D}_t}}{\sigma_{\hat{D}_t}} \right)^2. \tag{6}$$

where $D_t$ is the predicted depth prior, $\sigma_{D_t}$ and $u_{D_t}$ denotes the standard deviation and means respectively. Since it is based on the input, The second term is a constant. For the third term, we substitute the $\sigma_{\hat{D}_t}$ using its definition: $\sigma_{\hat{D}_t} = \sqrt{\frac{\sum_{i=1}^{N}\left( \hat{D}_t^i - u_{\hat{D}_t} \right)^2}{N}}$. The sum of the third term over all pixels in a depth map is: $\Sigma_i^N (\left( \frac{\hat{D}_t^i - u_{\hat{D}_t}}{\sigma_{\hat{D}_t}} \right)^2) = N$, Where $N$ is the total number of pixels.

We can see that both the second term and third term are constant. The first term is a simple transformation of the Pearson correlation coefficient. Thus, minimizing the $L2$ distance is equivalent to maximizing the coefficient.

### A.6 RESULTS ON THE DAVIS DATASET

#### A.6.1 PREPROCESSING PROCEDURE

On the Davis dataset, we adopt the preprocessing procedure outlined in the concurrent method, Shape-of-Motion (Wang et al., 2024b), to obtain these camera poses. Specifically, we first run UniDepth (Piccinelli et al., 2024) to acquire camera intrinsics and depth maps. Then, we utilize a SLAM method, Droid-SLAM (Teed & Deng, 2021), with the depth maps from UniDepth as input to solve for the camera poses.

#### A.6.2 COMPARISON WITH SHAPE-OF-MOTION

We provide an experiment to compare Shape-of-Motion with our method. In our experiments, we train shape-of-motion on the Davis dataset following their default training setting and use the same preprocessing strategy like depth estimation to train MoDGS. The visual comparison is shown in Fig. 11, which demonstrates that our method shows comparable results as Shape-of-Motion but renders more details in some cases. We also present the quantitative comparison results on the Nvidia dataset (Yoon et al., 2020) in Table 3.

Table 3: Comparison with Shape-of-Motion(SoM) on Nvidia dataset

| Method | Balloon2-2 | | | Jumping | | | Truck-2 | | | Skatting2 | | |
| --- | --- | --- | --- | --- | --- | --- | --- | --- | --- | --- | --- | --- |
| | PSNR | SSIM | LPIPS | PSNR | SSIM | LPIPS | PSNR | SSIM | LPIPS | PSNR | SSIM | LPIPS |
| SoM | 19.52 | 0.5096 | 0.2790 | 20.32 | 0.6771 | 0.2560 | 22.53 | 0.7414 | 0.1853 | 23.51 | 0.7861 | 0.1658 |
| Ours | **20.47** | **0.5275** | **0.2408** | **22.18** | **0.7075** | **0.2125** | **23.69** | **0.7455** | **0.1551** | **25.64** | **0.7996** | **0.1518** |

### A.7 DISCUSSION ABOUT ALPHA VALUE IN ORDINAL DEPTH LOSS

After roughly aligning the depth maps, we normalize the scene to $[-1, 1]^3$ and empirically set alpha to 100. We present further ablations in Table 4, where decreasing alpha leads to worse results while increasing alpha yields comparable results. The reason is that large alpha values compel the loss to mainly concentrate on the ordinality rather than the differences in depth values.

### A.8 DISCUSSION ABOUT THE ROBUSTNESS TO THE DEPTH NOISES

To show the robustness of our ordinal depth loss to depth noise, we manually add Gaussian noises to depth maps and use the noisy depth maps as supervision signals. The metric results of them are

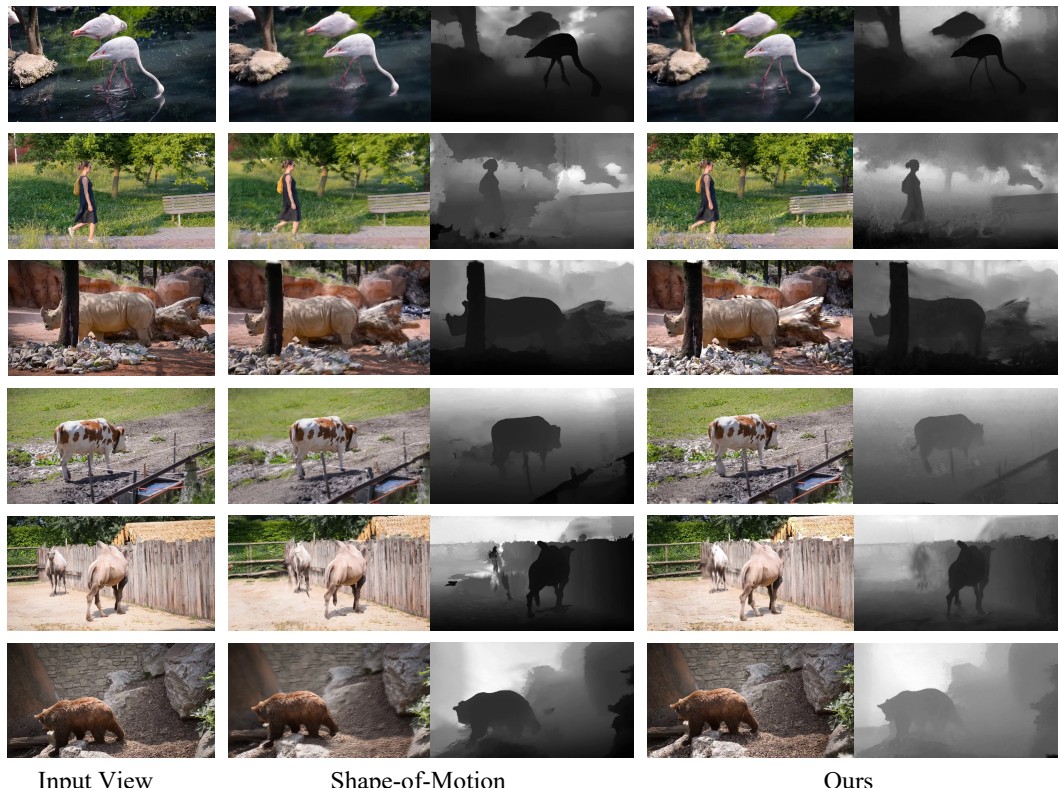

Input View         Shape-of-Motion         Ours

Figure 11: We show a visual comparison of rendered novel view images and depth maps between Shape-of-Motion(Wang et al., 2024b) and MoDGS. The rendering quality is comparable but our method shows more details in some cases.

Table 4: Quantitative results for different alpha values.

| Setting | PSNR | SSIM | LPIPS |
|---|---|---|---|
| alpha=10 | 21.05 | 0.7575 | 0.2370 |
| alpha=40 | 21.39 | 0.7813 | 0.1916 |
| alpha=100(Ours) | 23.23 | 0.8083 | 0.1592 |
| alpha=1000 | 23.16 | 0.8058 | 0.1715 |

shown in Table 5 (on the "flame_steak" scene), which shows our method is robust to small depth noises.

Table 5: Quantitative results with different levels of added Gaussian noise $Norm(\mu, \sigma)$

| Setting | PSNR | SSIM | LPIPS |
|---|---|---|---|
| $Norm(0, 0.20)$ | 23.18 | 0.8090 | 0.1667 |
| $Norm(0, 0.05)$ | 23.35 | 0.8100 | 0.1633 |
| Ours | 23.23 | 0.8083 | 0.1592 |

## A.9 PER-SCENE BREAKDOWN RESULTS ON THE DYNERF (LI ET AL., 2022) DATASET AND THE NVIDIA (YOON ET AL., 2020) DATASET

In Tables 6 and Table 7, we provide a breakdown of the metrics for different scenes in the DyNeRF and Nvidia datasets, respectively.

Table 6: Per-scene results on the DyNeRF dataset.

| Method | cut_roasted_beef | | | sear_steak | | | coffee_martini | | |
|---|---|---|---|---|---|---|---|---|---|
| | PSNR | SSIM | LPIPS | PSNR | SSIM | LPIPS | PSNR | SSIM | LPIPS |
| **HexPlane** | 16.76 | 0.5382 | 0.5054 | 16.89 | 0.5897 | 0.5049 | 13.26 | 0.4049 | 0.5835 |
| **SC-GS** | 20.69 | 0.7414 | 0.2625 | 21.23 | 0.7870 | 0.2188 | 19.02 | 0.7124 | 0.2151 |
| **D-GS** | 22.20 | 0.7808 | 0.1931 | **23.56** | 0.8101 | 0.1773 | 19.23 | 0.7013 | 0.2270 |
| **Ours** | **23.98** | **0.8221** | **0.1438** | 23.53 | **0.8126** | **0.1642** | **21.37** | **0.7962** | **0.1473** |

| Method | cook_spinach | | | flame_steak | | | flame_salmon_1 | | |
|---|---|---|---|---|---|---|---|---|---|
| | PSNR | SSIM | LPIPS | PSNR | SSIM | LPIPS | PSNR | SSIM | LPIPS |
| **HexPlane** | 16.95 | 0.7286 | 0.2223 | 16.97 | 0.7528 | 0.2543 | 11.16 | 0.3417 | 0.6382 |
| **SC-GS** | 16.70 | 0.7377 | 0.2117 | 17.31 | 0.7532 | 0.2527 | 17.65 | 0.6834 | 0.2253 |
| **D-GS** | 17.20 | 0.7195 | 0.2329 | 16.62 | 0.7523 | 0.2559 | 18.48 | 0.7038 | 0.2166 |
| **Ours** | **22.40** | **0.7823** | **0.1728** | **23.23** | **0.8083** | **0.1592** | **21.33** | **0.8038** | **0.1399** |

Table 7: Per-scene results on the Nvidia dataset.

| Method | Playground | | | DynamicFace-2 | | | Balloon1-2 | | | Umbrella | | |
|---|---|---|---|---|---|---|---|---|---|---|---|---|
| | PSNR | SSIM | LPIPS | PSNR | SSIM | LPIPS | PSNR | SSIM | LPIPS | PSNR | SSIM | LPIPS |
| HexPlane | 13.30 | 0.1641 | 0.5240 | 9.92 | 0.1873 | 0.6091 | 15.48 | 0.2793 | 0.5342 | 18.70 | 0.2238 | 0.4427 |
| SC-GS | 12.60 | 0.2059 | 0.5063 | 9.75 | 0.3312 | 0.4307 | 16.63 | **0.4412** | 0.3656 | 18.71 | **0.3501** | 0.3663 |
| D-GS | 12.62 | 0.2009 | 0.5116 | 10.36 | 0.3277 | 0.4563 | 10.36 | 0.3277 | 0.4563 | 19.16 | 0.3381 | 0.3300 |
| Ours | **13.38** | **0.2343** | **0.4542** | **12.40** | **0.4118** | **0.2442** | **16.93** | 0.4371 | **0.3059** | **19.47** | 0.3246 | **0.3004** |

| Method | Balloon2-2 | | | Skatting2 | | | Truck-2 | | | Jumping | | |
|---|---|---|---|---|---|---|---|---|---|---|---|---|
| | PSNR | SSIM | LPIPS | PSNR | SSIM | LPIPS | PSNR | SSIM | LPIPS | PSNR | SSIM | LPIPS |
| HexPlane | 18.42 | 0.3159 | 0.4881 | 21.39 | 0.6403 | 0.3983 | 20.85 | 0.5687 | 0.3653 | 19.28 | 0.5605 | 0.4431 |
| SC-GS | 18.28 | 0.3893 | 0.3671 | 23.07 | 0.7186 | 0.2002 | 21.48 | 0.6374 | 0.2078 | 20.17 | 0.6698 | 0.2343 |
| D-GS | 18.57 | 0.3941 | 0.3768 | 24.47 | 0.7582 | 0.2073 | 21.38 | 0.6321 | 0.2142 | 21.27 | 0.6604 | 0.2700 |
| Ours | **20.47** | **0.5275** | **0.2408** | **25.64** | **0.7996** | **0.1518** | **23.69** | **0.7455** | **0.1551** | **22.18** | **0.7075** | **0.2125** |

## A.10 COMPARISON WITH DEPTH RANKING LOSS

We provide a quantitative comparison between our ordinal depth loss and other depth ranking losses (Chen et al., 2016; Pavlakos et al., 2018) in Table 8. In this experiment, we replace our ordinal depth loss with their depth ranking loss while keeping all other configurations fixed. The results demonstrate that our ordinal depth loss significantly outperforms their approach across all scenes. This highlights the effectiveness of our method in capturing depth information more accurately than the existing depth ranking losses.

In Fig. 12, we present a visual comparison of rendered depth maps using depth ranking loss and ordinal depth loss. Our depth values are more smooth and well-distributed while depth values learned from rank depth loss show high contrast and more noise. The reason is that depth ranking loss encourages large contrast to satisfy the given depth order and is not robust to the noises in the input depth.

Table 8: Comparison with depth ranking loss (Chen et al., 2016; Pavlakos et al., 2018), denoted as $DRL$ for simplification.

| | sear_steak | | | cut_beef | | | coffee_martini | | |
|---|---|---|---|---|---|---|---|---|---|
| | PSNR | SSIM | LPIPS | PSNR | SSIM | LPIPS | PSNR | SSIM | LPIPS |
| $DRL$ | 18.22 | 0.7390 | 0.2228 | 17.07 | 0.7162 | 0.2283 | 14.09 | 0.7465 | 0.1902 |
| Ours | **23.53** | **0.8126** | **0.1642** | **23.98** | **0.8221** | **0.1438** | **21.37** | **0.7962** | **0.1473** |

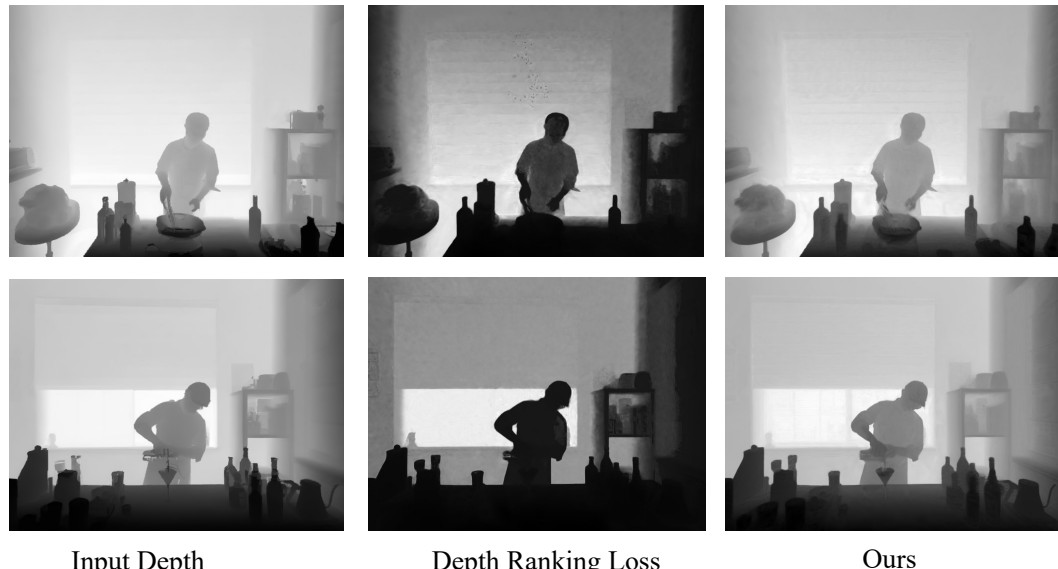

|          Input Depth          |     Depth Ranking Loss     |          Ours          |

Figure 12: A visual comparison of rendered depth maps between using depth ranking loss (Wang et al., 2024b) and our ordinal depth loss.

## A.11 ABLATION USING DEPTH FROM VIDEO-DEPTH ESTIMATION METHODS

To show that our method is also compatible with recent video depth methods, we conduct an ablation experiment using the depth maps predicted by the video depth estimator (Shao et al., 2024) and different types of depth loss as supervision. As shown in Table 9, the video depth estimator improves the results, and our ordinal depth loss still consistently improves the results.

Table 9: Video depth ablation. $VD + P$ refers to supervised using video depth maps with pearson depth loss, $VD + O$ refers to supervised using video depth maps with ordinal depth loss.

|          | sear_steak | | | cut_beef | | | coffee_martini | | |
|----------|------|------|-------|------|------|-------|------|------|-------|
|          | PSNR | SSIM | LPIPS | PSNR | SSIM | LPIPS | PSNR | SSIM | LPIPS |
| $VD + P$ | 23.43 | 0.7983 | 0.1732 | 23.19 | 0.8134 | 0.1472 | 20.01 | 0.7664 | 0.1647 |
| $VD + O$ | 23.48 | 0.8091 | 0.1711 | 23.61 | 0.8103 | 0.1532 | 21.55 | 0.8001 | 0.1488 |

## A.12 RESULTS OF NERF-BASED METHODS USING DEPTH SUPERVISION

We present the results of HexPlane (Cao & Johnson, 2023) when supervised with depth loss in Table 10. Using depth maps for supervision only leads to a slight improvement for HexPlane and MoDGS still outperforms HexPlane by a large margin.

Table 10: Metric results when HexPlane (Cao & Johnson, 2023) is incorporated with depth supervision, denoted as "HexPlane+D".

|            | flame_steak | | | cut_beef | | | coffee_martini | | |
|------------|------|------|-------|------|------|-------|------|------|-------|
|            | PSNR | SSIM | LPIPS | PSNR | SSIM | LPIPS | PSNR | SSIM | LPIPS |
| HexPlane   | 16.97 | 0.7528 | 0.2543 | 16.76 | 0.5382 | 0.5054 | 13.26 | 0.4049 | 0.5835 |
| HexPlane+D | 17.21 | 0.6156 | 0.4646 | 18.82 | 0.6344 | 0.414 | 15.78 | 0.4708 | 0.5106 |
| Ours       | **23.23** | **0.8083** | **0.1592** | **23.98** | **0.8221** | **0.1438** | **21.37** | **0.7962** | **0.1473** |

## A.13 DIFFERENT RANDOM INITIALIZATION METHODS

In the ablation study of our paper, we use the Kaiming initialization as the random initialization. We additionally provide ablations using zero deformation initialization of the deformation field in Table 11(flame_steak scene of DyNeRF). In this experiment, we still initialize the canonical Gaussian spaces but adopt a different initialization strategy for the deformation field MLP.

Table 11: Quantitative comparison results for different initialization methods

| Method | PSNR | SSIM | Lpips |
|---|---|---|---|
| ZeroDeformationInit | 22.34 | 0.7906 | 0.2151 |
| KaimingInit | 22.24 | 0.7744 | 0.2036 |
| 3D-AwareInit (Ours) | **23.23** | **0.8083** | **0.1592** |

## A.14 LENGTH OF VIDEOS THAT MODGS CAN HANDLE

Currently, we can handle videos up to 10 seconds. Scaling to longer videos(~30 seconds) is possible if we use a larger deformation MLP network and train for more iterations.

## A.15 DISCUSSION ABOUT DEPTH CONSISTENCY METHODS

From the knowledge distillation perspective, our method can be regarded as a distillation framework to get consistent video depth. In our framework, we adopt the 3D Gaussian field as the 3D representation and the ordinal depth loss as the supervision to distill the monocular depth estimation. Meanwhile, we adopt the rendering loss to further regularize the 3D representation, which enables us to distill temporally consistent monocular video depth from the inconsistent estimated depth maps. The inconsistent input depth and the distilled consistent video depth can be visualized in the supplementary video. This depth distillation mechanism is also adopted by self-supervised video depth estimation methods (Wang et al., 2023d; 2024c; Xu et al., 2024; Luo et al., 2020).

Although our method primarily targets novel-view synthesis of dynamic scenes, we also conducted an experiment to compare the depth map quality of MoDGS with a rencent video depth stabilization method (Wang et al., 2023d). In Fig. 16, it can be observed that the depth maps from MoDGS contain more details and are more temporally stable. We sincerely recommend you to watch our supplementary videos.

## A.16 COMPARISON WITH PERCEPTUAL DEPTH LOSS

In this experiment, we substitute our ordinal depth loss with a perceptual depth loss, specifically the learned perceptual image patch similarity loss (LPIPS) (Zhang et al., 2018) while maintaining all other experimental conditions unchanged. For the LPIPS loss, we utilize the AlexNet (Krizhevsky et al., 2012) in computing the LPIPS loss. We present a quantitative comparison of our ordinal depth loss against perceptual depth losses in Table 12. The results indicate that our ordinal depth loss outperforms the alternative LPIPS depth loss. Furthermore, we provide a visual comparison of the rendered depth maps using perceptual losses in Fig. 13. Compared to the maps produced with the LPIPS loss, our depth maps are smoother and contain less noise.

Table 12: Comparison between perceptual depth loss ("LPIPS") and our original depth loss ("Ordinal").

| | sear_steak | | | cut_beef | | | coffee_martini | | |
|---|---|---|---|---|---|---|---|---|---|
| | PSNR | SSIM | LPIPS | PSNR | SSIM | LPIPS | PSNR | SSIM | LPIPS |
| **LPIPS** | 22.43 | 0.8049 | 0.1782 | 23.36 | 0.8102 | 0.1591 | 20.97 | 0.7820 | 0.1575 |
| **Ordinal** | **23.53** | **0.8126** | **0.1642** | **23.98** | **0.8221** | **0.1438** | **21.37** | **0.7962** | **0.1473** |

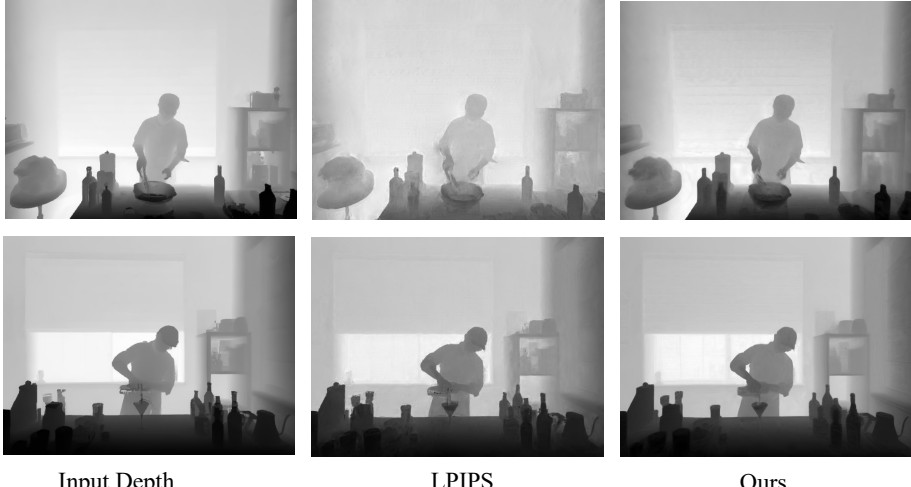

| Input Depth | LPIPS | Ours |

Figure 13: A visual comparison of rendered depth maps using the perceptual depth loss and our ordinal depth loss.

## A.17 RESULTS ON SCENES WITH HIGHLY COMPLEX MOTIONS

Reconstructing 4D dynamic fields in complex scenes using casually captured monocular videos remains a challenging task. In $playground$ and $balloon2-2$ scenes, which both contain two moving objects (the human and balloon) and complex motion, MoDGS handles the situation relatively well and generates reasonable results. We present visual results in Fig. 15. As shown in the figure, the appearances of both objects can be well reconstructed. For more complex motions, MoDGS may fail due to the difficulty in predicting robust depth maps, which we leave for future works.

## A.18 RESULTS ON SCENES WITH RAPID MOTION

Rapidly moving scenes pose significant challenges for dynamic Gaussian field reconstruction. Such rapid motions can lead to inaccurate camera pose estimations, which remain problematic for MoDGS and all existing dynamic Gaussian reconstruction methods. We conducted an experiment on a high-speed motion scene($rollerblade$ from the Davis dataset) containing motion blur (in the regions of hands, feet, and hair) to evaluate our MoDGS performance under challenging conditions. As shown in Fig. 15, MoDGS produces reasonable results in the severely blurred regions. The results demonstrate that our approach maintains robust functionality even in rapidly changing environments. To better address this issue, we may combine deblur methods (Zhu et al., 2023b;a; Pan et al., 2020) with motion priors in future works.

## A.19 DISCUSSION ON HOW TO HANDLE SCENARIOS WITH HEAVY OCCLUSIONS OR SPECULAR.

**How to handle scenarios with heavy occlusions.**  If the occluded regions are observed at some timestamps, MoDGS is able to accumulate information among different timestamps to complete these occluded regions on some timesteps as shown in Appendix A.4. However, MoDGS cannot generate new contents for the occluded regions, which indeed degenerate the rendering quality on occluded regions. Incorporating some generative priors such as diffusion models could alleviate this problem.

**Possible solutions dealing with specular scenarios.**  MoDGS can reconstruct some specular objects but show some artifacts because MoDGS does not involve any special design for specular objects. As shown by a recent work (Fan et al., 2024), a potential solution for this is to introduce inverse rendering in dynamic scenes, which enables accurate modeling of specular surfaces in dynamic scenes.

A.20    DISCUSSION ABOUT THE ROBUSTNESS TO DIFFERENT DEPTH ESTIMATORS

To further demonstrate the robustness of our ordinal depth loss across different depth estimation methods, we compare the performance of MoDGS using various depth estimators (Yang et al., 2024a; Ke et al., 2023; Shao et al., 2024) while keeping all other configurations unchanged. The quantitative results obtained using these methods are presented in Table 13. When different depth maps are utilized, the performance of our method exhibits minor variation, which demonstrates that our method is robust to various depth estimation approaches and does not rely on a specific depth prior distribution, such as Geowizard (Fu et al., 2024).

In Fig. 17, We also provide a visual comparison of depth maps generated by these methods and rendered using our MoDGS. Although some flickering and inconsistencies can be observed in the input depth maps, MoDGS still produces stable and consistent depth renderings. We sincerely recommend you to watch our supplementary videos.

Table 13: Quantitative results with different depth estimation methods, such as Marigold (Ke et al., 2023), ChronoDepth (Shao et al., 2024), DepthAnything (Yang et al., 2024a) and GeoWizard (Fu et al., 2024).

| Depth estimator | PSNR | SSIM | LPIPS |
|---|---|---|---|
| Marigold (Ke et al., 2023) | 23.12 | 0.8057 | 0.1712 |
| ChronoDepth (Shao et al., 2024) | 23.44 | 0.8105 | 0.1687 |
| DepthAnything (Yang et al., 2024a) | 23.35 | 0.8097 | 0.1621 |
| GeoWizard (Fu et al., 2024) | 23.23 | 0.8083 | 0.1592 |

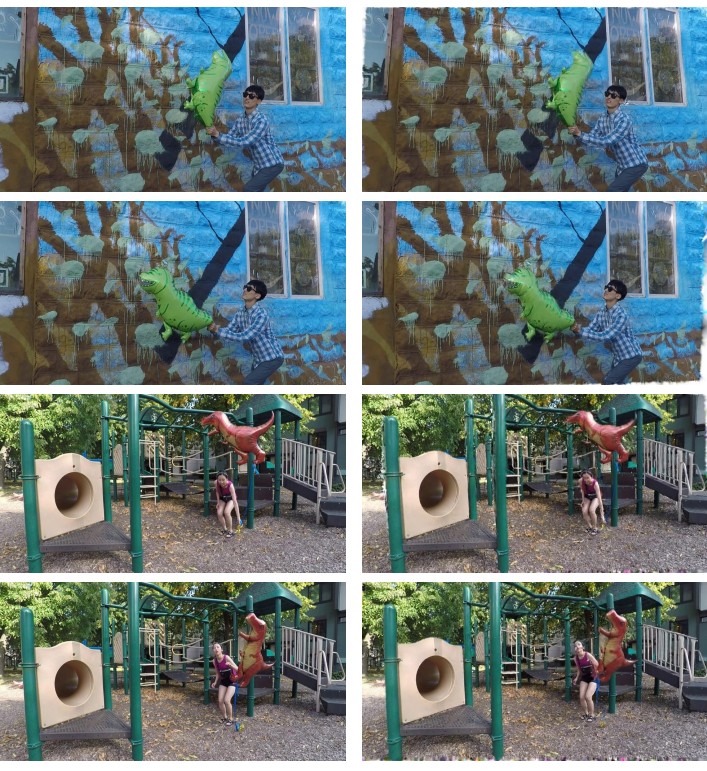

Input View                                    Rendered View

Figure 14: Input views and novel view synthesis results from MoDGS for two scenes with complex motions in the NVIDIA dataset.

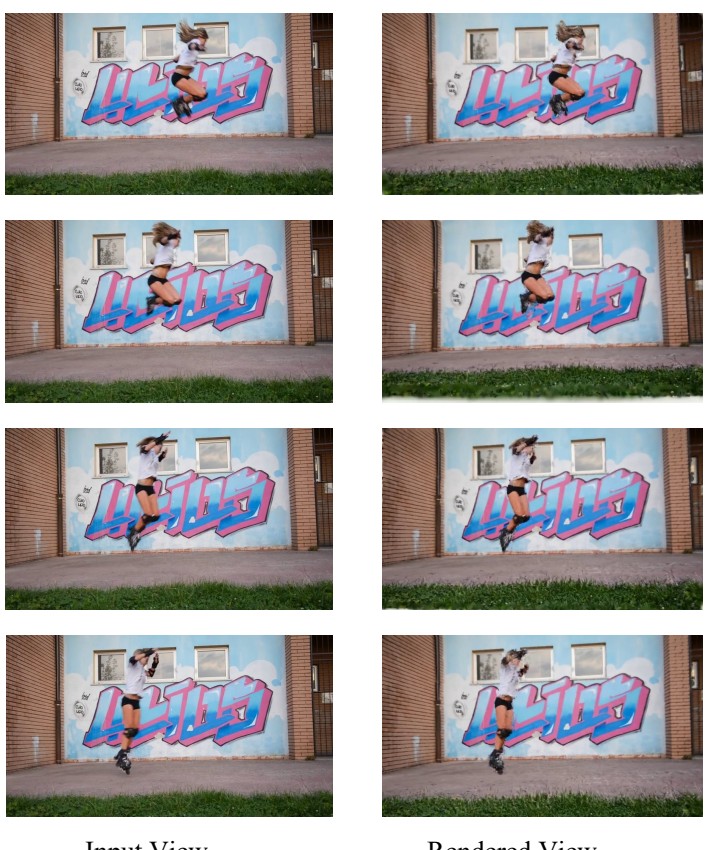

Input View                                 Rendered View

Figure 15: Input views and novel view synthesis results produced by MoDGS on the $rollerblade$ scene from the Davis dataset, which features rapid motions and motion blur.

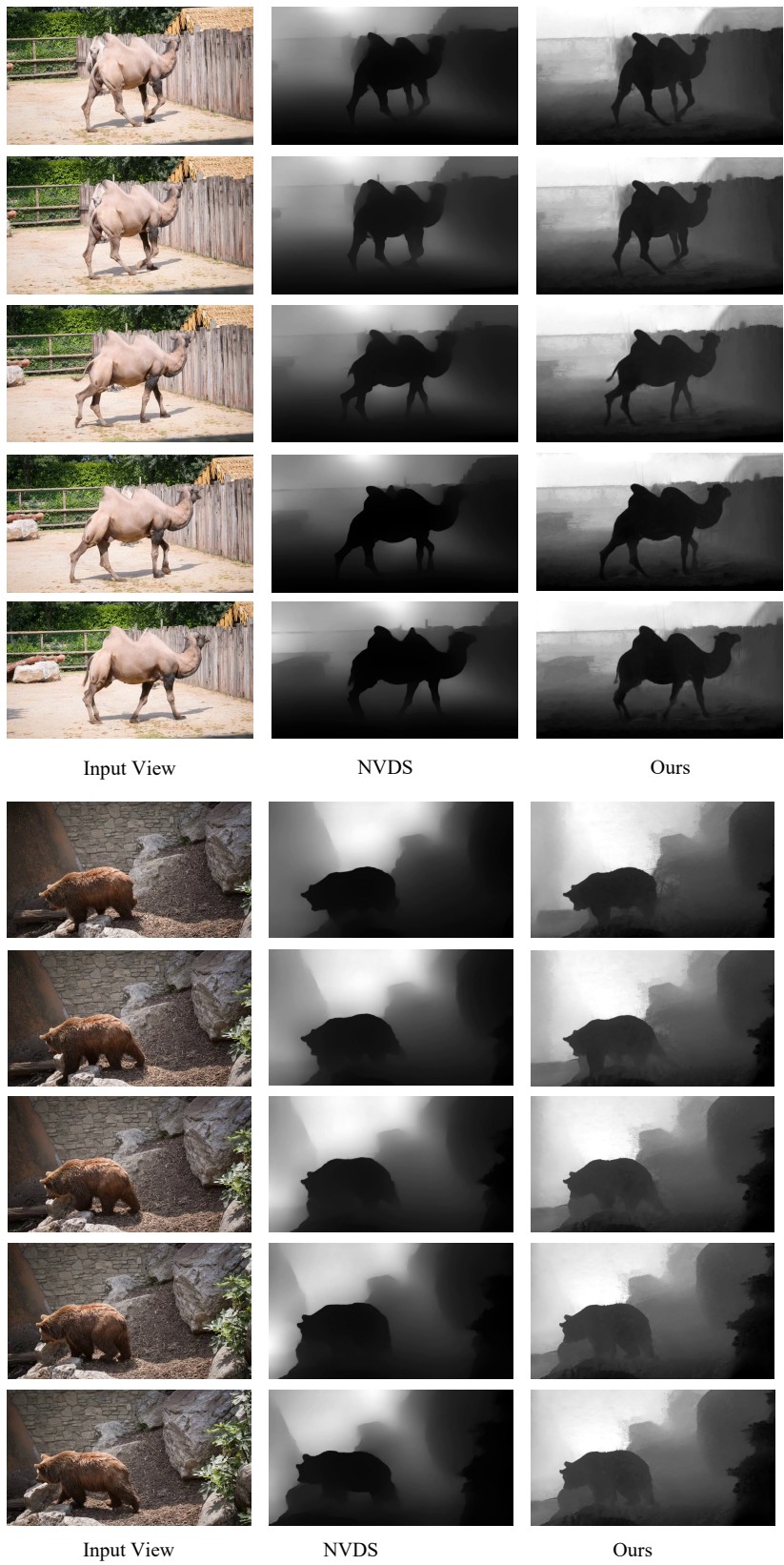

Figure 16: A visual comparison of depth maps estimated by NVDS (Wang et al., 2023d) and rendered by our MoDGS on $Camel$ and $Bear$ scene from the Davis dataset.

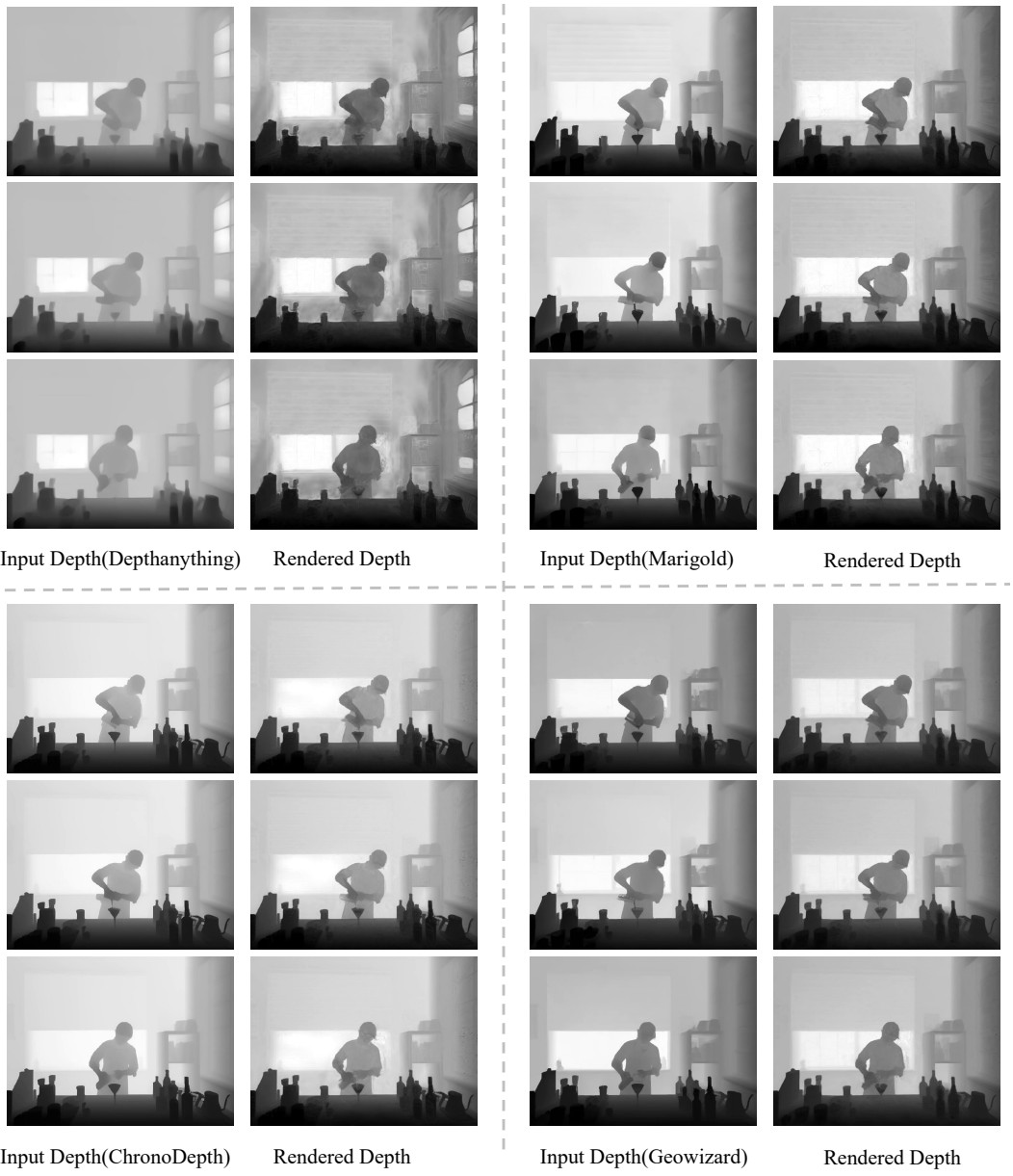

Figure 17: A visual comparison of rendered depth maps from our MoDGS and input depth maps estimated by different depth estimation methods, such as Marigold (Ke et al., 2023), ChronoDepth (Shao et al., 2024), DepthAnything (Yang et al., 2024a) and GeoWizard (Fu et al., 2024).

