# OpenReview forum: "MoDGS: Dynamic Gaussian Splatting from Casually-captured Monocular Videos with Depth Priors"
_ICLR.cc/2025/Conference — ICLR 2025 Poster_

### Official Review · Reviewer_Gzks · 2024-10-31

**Soundness:** 2
**Presentation:** 3
**Contribution:** 2
**Rating:** 6
**Confidence:** 4

**Summary:**

The paper introduces MoDGS, a method for dynamic view synthesis from monocular videos, leveraging a Gaussian-based splatting technique combined with deformation fields and an ordinal depth loss to reconstruct scenes. This framework integrates a 3D-aware initialization to align Gaussian representations in a canonical space, while the ordinal depth loss is used to improve scene geometry continuity. MoDGS is claimed to improve over previous dynamic NeRF approaches and related deformation methods, with results evaluated on the DyNeRF and Nvidia datasets.

**Strengths:**

1. I think the 3D-aware initialization process is a strong point, as it specifically addresses a common issue in monocular reconstruction. By initializing Gaussians instead of relying on random initialization, this method seems to potentially add more consistency.

2. The ordinal depth loss is, in my view, an interesting idea. It tries to tackle scale ambiguity in monocular depth estimation, which I think is particularly relevant in dynamic scenes. This loss formulation promotes depth consistency across frames, an essential factor when handling complex, moving scenes.

**Weaknesses:**

1. I think the innovation is quite incremental for the reason that compared to closely related works like Deformable 3DGS and 4DGS, the methodological innovation appears incremental, mainly optimizing existing elements (depth consistency and deformation) rather than proposing a new structural approach.


2. Besides, the approach relies heavily on pre-trained depth models. MoDGS relies on single-view depth estimators like GeoWizard for depth initialization, which brings into question the independence of its results. The approach leverages external models as priors, potentially limiting its novelty and raising questions regarding knowledge distillation. The extent to which these pre-trained models influence the final performance is not rigorously analyzed.


3. While MoDGS integrates external depth estimation for initialization, there is no formalized knowledge distillation to adaptively refine the model during training. This absence may reduce the adaptability of MoDGS across different dynamic scenes where pre-trained depth estimators may not perform equally well.

**Questions:**

1. How does MoDGS handle scenarios where the pre-trained depth estimator provides inconsistent depth due to environmental variations? Has any analysis been conducted to measure performance stability when GeoWizard or other models are less reliable?


2. Would MoDGS perform as well on datasets with higher motion complexity or less predictable scene geometry? Testing on a broader range of datasets, such as those with cluttered backgrounds or multiple moving objects, would better validate the method's generalization.

3.  Considering MoDGS’s reliance on single-view depth priors, would a formalized knowledge distillation framework improve model autonomy by adapting these priors dynamically during training?

---

> ### Author Response · Authors · 2024-11-21
>
> Thank you for your helpful comments! Our replies to your questions and revisions to the submission are stated below.
> ## Weakness
>
> ### W1: The innovation is quite incremental in comparison with closely related works like Deformable 3DGS and 4DGS.
> ### A1:
>
> - **Difference between our MoDGS and baseline methods, Deformable 3DGS and 4DGS**.
> As introduced in L036, Deformable-3DGS and 4DGS require multiview videos or "teleported" monocular videos as inputs, which is not a real monocular video setting.
> In contrast, MoDGS is able to reconstruct 4D fields from real **casually** captured monocular videos, which move smoothly and slowly and thus are significantly more challenging.
> Deformable-3DGS and 4DGS fail on these casually captured monocular videos as demonstrated in our experiments including videos (from 0m12s to 0m56s) and tables (Tab.1, Tab.6, and Tab.7).
> - **Our contributions**. MoDGS contains two novel techniques, i.e. 3D-aware-init and ordinal depth loss.
> 1) Gaussian splatting can be initialized from SfM points in static scenes but how to robustly initialize dynamic Gaussian fields is not well-studied.
> We show that the previous initialization method adopted in dynamic 3DGS does not work well in the real monocular setting in our experiments(Fig.7 and Tab.2).
> Thus, we propose 3D-aware-init which provides a robust starting point for the deformation field and canonical space Gaussians.
> 2) In the real monocular setting, we have to rely on monocular depth estimation due to the weak multiview constraints in casually captured videos. However, how to supervise the dynamic Gaussian fields with inaccurate monocular depth maps is still an open question.
> The proposed ordinal depth loss is designed to effectively utilize the inconsistent depth for the 4D reconstruction.
>
>
> ### W2: The approach relies heavily on pre-trained depth models. The approach leverages external models as priors, potentially limiting its novelty and raising questions regarding knowledge distillation. The extent to which these pre-trained models influence the final performance is not rigorously analyzed.
> ### A2:
> - **Why rely on monocular depth?**
> As introduced in L041, casually captured videos usually have minor camera motions, which provide insufficient multiview constraints. Thus, we have to introduce monocular depth estimators to constrain the geometry, which is also adopted by all concurrent works[1,2,3,4] that process casual monocular video.
>
> - **Is the proposed method robust to the noise of estimated depth?** In Appendix A.8, we present an ablation study on robustness to depth quality by introducing Gaussian noise into the input depth maps. After adding Gaussian noise, the PSNR changed by less than 0.2, the SSIM by less than 0.008, and the LPIPS by less than 0.002. The results demonstrate that our method is robust to a certain range of noise and is not sensitive to variations in depth quality.
>
>
> - **Emerging trends in utilizing off-the-shelf models for specialized tasks.** Last but not least, it is worth noting that in the era of deep learning, the performance of large models tailored for specific tasks has significantly improved due to the surge in data and computing power. Utilizing pre-trained off-the-shelf models as one component of a multi-stage processing system to be constructed is becoming a prominent and emerging trend.
>
>
>
> ### W3: While MoDGS integrates external depth estimation for initialization, there is no formalized knowledge distillation to adaptively refine the model during training. This absence may reduce the adaptability of MoDGS across different dynamic scenes where pre-trained depth estimators may not perform equally well.
> ### A3:
> - **How do we distill the monocular depth for supervision?** We adopt the ordinal depth loss as the supervision loss to constrain the geometry from the estimated monocular depth maps.
> - **Is MoDGS robust to different depth estimation methods or noises in the depth estimation?** Yes. The table above has proved the robustness to noise. We have also tested our method in combination with different depth techniques in Tab. 9, all of which yielded effective results compared with Pearson depth loss.
>
> - **Why our method is robust to noise in the depth estimation?** As introduced in L305,  the estimated depth maintains relatively stable orders, even though the absolute values can vary significantly. Our ordinal depth loss relies solely on the orders of pixel pairs. Moreover, our rendering loss will provide further corrective effects.

---

> > ### Comment · Reviewer_Gzks · 2024-11-22
> >
> > Thank you for addressing the concerns raised in the review.
> >
> > ### **Regarding W1: Incremental Innovation**
> >
> > Thank you for providing detailed clarification on MoDGS's contributions. I agree that the 3D-aware-init technique is particularly interesting. It addresses a notable gap in initializing dynamic Gaussian fields in the real monocular setting.
> >
> > As I mentioned earlier, I find the overall framework of the paper to lean toward being incremental in comparison with related works. However, the inclusion of 3D-aware-init stands out as a systemically innovative contribution. I agree with your statement that initialization significantly impacts the final results, and 3D-aware-init demonstrates notable practical value in addressing the challenges of robustly initializing dynamic Gaussian fields. Thank you for your clarification.
> >
> >
> > ### **Regarding W2: Reliance on Pre-trained Depth Models**
> > I acknowledge that using monocular depth estimators as a necessity in weak multiview settings is understandable and aligns with common practices in the field not only NVS but 3D detection/segmentation/etc. Your robustness tests (e.g., adding Gaussian noise) are appreciated and address some of the concerns. However:
> > - While robustness to noise is demonstrated, how do these results generalize to real-world depth errors or systematic biases in monocular depth estimators?  Real-world errors in monocular depth estimation typically exhibit the following characteristics:
> >
> >     - **Systematic Biases**: Real-world errors often exhibit consistent biases, such as scale drift or depth compression, which accumulate over time and cannot be modeled by random noise.
> >
> >     - **Spatial/Temporal Correlations**: Depth errors are often spatially structured or temporally inconsistent, unlike the pixel-independent nature of Gaussian noise.
> >
> >     - **Scene Dependency**: Errors vary by scene complexity (e.g., occlusions, dynamic objects), introducing challenges not captured by Gaussian noise.
> >
> > To strengthen the evaluation, qualitative visualizations of real-world depth errors and experiments with **multiple pre-trained depth models (highlighting common biases)** would provide a more realistic assessment of MoDGS's robustness.
> >
> > Overall， the authors' rebuttal has given me a new perspective on the systemic innovations in this work, particularly the practical and impactful contributions of 3D-aware-init. This insight has addressed my initial concerns about the incremental nature of the paper and highlighted its value in advancing the field.
> > As a result, I will raise my score to reflect this improved understanding of the work's contributions, I will further adjust my score if further convinced.
> >
> >
> > ### **Regarding W3: Knowledge Distillation**.
> >
> > Your rebuttal highlights the use of ordinal depth loss and the reliance on stable ordinal relationships rather than absolute depth values. This is a thoughtful approach and mitigates some of the challenges of monocular depth estimation.   However, the same concerns are raised as W2.
> >
> > ---
> >
> > **Overall**, the authors' rebuttal has given me a new perspective on the systemic innovations in this work, particularly the practical and impactful contributions of 3D-aware-init. This insight has addressed my initial concerns about the incremental nature of the paper and highlighted its value in advancing the field.
> >
> > As a result, I will raise my score to reflect this improved understanding of the work's contributions, I will further adjust my score if further convinced.

---

> ### Author Response · Authors · 2024-11-21
>
> ### Q1: How does MoDGS handle scenarios where the pre-trained depth estimator provides inconsistent depth due to environmental variations? Has any analysis been conducted to measure performance stability when GeoWizard or other models are less reliable?
>
> ### A1:
> - The depth inputs indeed exhibit scale inconsistencies, and we did not assume these would be consistent as shown in the input depth videos of the supplementary demo video)
>
> - In the 3D-aware-init stage, we address this inconsistency by estimating a rough scale for each depth map (see Sec 3.2, in *Initialization of depth scales* paragraph), which does not need to be very accurate.
> - In the optimization stage, we did not directly apply L1 or Pearson correlation depth loss for direct supervision but adopted the ordinal depth loss because we observed that the depth orders are relatively more consistent.
>
>
> ### Q2: Would MoDGS perform as well on datasets with higher motion complexity or less predictable scene geometry? Testing on a broader range of datasets, such as those with cluttered backgrounds or multiple moving objects, would better validate the method's generalization.
> ### A2:
> - Scenes such as *deaddrift*(from MCV dataset, results in Fig.6 and video starts at 3m20s), *coffee_martini*(from DyNeRF dataset, results in Fig.5 and video starts at 2m12s), and  *camel*(from Davis dataset, results in Fig.11 and video starts at 3m44 ) are challenging due to their complex movements and cluttered backgrounds. Nevertheless, our method remains effective and produces reasonable results in these scenarios.
> - In the *playground* and *balloon2-2* scenes, which feature two moving objects (the person and the balloon) and complex motion, MoDGS handles the situation relatively well and generates reasonable results. In Appendix A.17, we present the NVS results on these two scenes and we will update the demo video in the final version.
> - We agree with the reviewer that reconstructing 4D dynamic fields in complex scenes with just casually captured monocular videos is still a challenging task. Our work already makes significant improvements in this setting with moderate complexity, which could be the basis for addressing more cluttered scenes with more complex motions in future works.
>
>
> ### Q3: Considering MoDGS’s reliance on single-view depth priors, would a formalized knowledge distillation framework improve model autonomy by adapting these priors dynamically during training?
>
> ### A3:
> We agree with the reviewer that our method can be regarded as a distillation framework from the perspective of knowledge distillation to get consistent video depth. In our framework, we adopt the 3D Gaussian field as the 3D representation and the ordinal depth loss as the supervision to distill the monocular depth estimation. Meanwhile, we adopt the rendering loss to further regularize the 3D representation, which enables us to distill temporally consistent monocular video depth from the inconsistent estimated depth maps. The inconsistent input depth and the distilled consistent video depth can be visualized in the supplementary video. We will add this discussion in the revision(Appendix A.15).
>
>
> - [1] Liu et al., Robust Dynamic Radiance Fields, CVPR, 2023.
> - [2] Lee et al., Fast View Synthesis of Casual Videos with Soup-of-Planes, ECCV, 2024.
> - [3] Lei et al., MoSca: Dynamic Gaussian Fusion from Casual Videos via 4D Motion Scaffolds, arXiv preprint, 2024.
> - [4] Wang et al., Shape of Motion: 4D Reconstruction from a Single Video, arXiv preprint arXiv:2407.13764, 2024.

---

> > ### Comment · Reviewer_Gzks · 2024-11-22
> >
> > Thank you for addressing the questions and providing additional insights into the method's robustness and potential. Below is my feedback on each point.
> >
> > The rebuttal provides strong justifications and additional insights that clarify key aspects of the method. While challenges remain in handling highly complex scenes and achieving greater autonomy through dynamic adaptation, the thoughtful responses and plans for revision have improved my understanding of the work's contributions. Thank you again.
> >
> > Providing the aforementioned additional analyses like“*qualitative visualizations of real-world depth errors and experiments with multiple pre-trained depth models (highlighting common biases)*” could further strengthen the claim of **systemic innovation and stability, distinguishing the method from one that relies on ad-hoc tuning to mitigate biases for one specific prior distribution(eg. one specific large-pre-train)**. I appreciate the effort in addressing the concerns and will reflect this in my final evaluation.

---

> > > ### Author Response · Authors · 2024-11-25
> > >
> > > Thank you very much for your prompt reply and insightful suggestions regarding experiments to demonstrate robustness.
> > >
> > > Following your advice, we conducted ablation studies using three additional depth estimation methods[1,2,3] and compared the performance of our MoDGS. The results are presented in Appendix A.20.  When we switched to different depth estimation methods, the PSNR varied by less than 0.35, the SSIM by less than 0.01, and the LPIPS by less than 0.015. These minimal changes demonstrate the robustness of our approach to varying distributions of real-world depth estimators. The main reason is that our method only relies on the order of depth values and most depth estimators are able to estimate correct depth orders in spite of their varying error types in different scenes. However, we also agree that the strong varying environments or cluttered scenes may break the correctness of predicted depth orders so our method may fail in these extremely challenging cases.
> > >
> > > Some qualitative visualization of depth inputs from different methods and depth maps rendered from MoDGS are added in manuscripts (A.20, Fig 17), and supplementary video (from  4m52s to 5m08s).  It can be observed that MoDGS produces consistent video depth from different types of estimated depth maps.
> > >
> > >
> > > - [1] Ke et al., Repurposing diffusion-based image generators for monocular depth estimation, CVPR, 2024.
> > > - [2] Yang et al., Depthanything: Unleashing the power of large-scale unlabeled data, CVPR, 2024.
> > > - [3] Shao et al., Learning Temporally Consistent Video Depth from Video Diffusion Priors, arXiv preprint, 2024.

---

> > > > ### Comment · Reviewer_Gzks · 2024-12-03
> > > >
> > > > Thank you for your feedback and additional experiments. Considering the fact that rendered images with different depth estimators somehow show consistency. I think it addresses my major concerns.
> > > >
> > > > I will raise my point to borderline acceptance. Good Luck

---

> ### Author Response · Authors · 2024-11-29
> **We are waiting for your further feedback. Thank you!**
>
> Dear Reviewer **Gzks**,
>
> We sincerely extend our gratitude for dedicating your time and effort to reviewing our submission to ICLR. Your acknowledgment of our efforts is greatly valued. In response to your further feedback, we have conducted ablation studies employing three supplementary depth estimation methods, showcasing the robustness of our approach. We kindly invite you to review our detailed responses at your earliest convenience and let us know if there are any remaining concerns.   We are eager to engage in further discussions on any outstanding issues.
>
> Thank you once again for your insightful feedback.
>
> Best regards,
>
> The authors

---

> ### Author Response · Authors · 2024-12-02
>
> Dear Reviewer **Gzks**,
>
> Thanks for your valuable time reviewing our paper. The discussion session is closing soon. We are eager to hear your additional feedback of our paper. Thank you!
>
> Best Regards,
>
> The Authors

---

> ### Author Response · Authors · 2024-12-03
>
> Dear Reviewer **Gzks**,
>
> Thank you very much for the insightful discussion and for taking the time to review our work. We truly appreciate your efforts in helping us enhance our paper. If you have any further questions,  we are more than happy to reply.
>
> Best regards,
>
> The Authors

---

### Official Review · Reviewer_FCsY · 2024-10-31

**Soundness:** 2
**Presentation:** 3
**Contribution:** 2
**Rating:** 5
**Confidence:** 4

**Summary:**

The paper presents MoDGS, a novel pipeline for synthesizing dynamic scenes from casually captured monocular videos. Unlike existing methods requiring large camera motions, MoDGS leverages single-view depth estimation for 3D reconstruction and introduces a 3D-aware initialization alongside an ordinal depth loss. These innovations enable robust, high-quality novel view synthesis, outperforming state-of-the-art methods in rendering casually captured videos.

**Strengths:**

* A differentiable order-based loss function, the ordinal depth loss, is proposed, with detailed descriptions of its motivation and its distinctions from other depth loss functions.
* It demonstrates significant superiority over multi-view camera methods in reconstruction metrics and visual results, with ablation studies validating the importance of the "3D-aware initialization scheme" and "ordinal depth loss."
* The paper is well-written and easy to follow.

**Weaknesses:**

* **The contributions and innovations are limited**. This work is based on the previous canonical space paradigm of 3D Gaussian Splatting (3DGS) combined with deformation fields, with the main contributions being a deformable 3DGS initialization method and a depth loss. The primary principle of the former relies on predicting per-pixel 3D flow using current state-of-the-art monocular depth estimation and optical flow estimation methods. However, the sole innovative aspect lies in converting 2D optical flow to 3D flow using the estimated depth map. As for the depth loss, although it is well-motivated and provides performance improvement, it essentially replaces the Pearson correlation loss with an order correlation loss.
* **The experimental comparisons lack fairness**. In most quantitative comparisons, this work is only compared against methods that require multi-view camera input. It is recommended to include quantitative and qualitative comparison results with methods under the same setting of "casually captured monocular video." It is also perplexing that the authors mention "RoDynRF, a method that adopts single-view depth estimation as supervision" in "Baseline methods", yet I only found comparative results for this method in Fig. 6.

**Questions:**

Kindly refer to the [Weaknesses].

---

> ### Author Response · Authors · 2024-11-21
>
> We thank the reviewer for the effort in reviewing our paper. Our response to the all concerns is listed below.
>
> ## Weakness
> ### W1: The contributions and innovations are limited because this work is based on the canonical space paradigm of 3DGS combined with deformation fields, with the main contributions being a deformable 3DGS initialization method and a depth loss.
> ### A1:
> - **Novelty in comparison with baseline methods with canonical space and deformation fields**.
> Although baseline methods Deformable-3DGS and 4DGS also use canonical space and deformation fields, these baseline methods all require multiview videos or "teleported" monocular videos as inputs, which is not a real monocular video setting.
> In contrast, our MoDGS is able to reconstruct 4D fields from real **casually** captured monocular videos, which move smoothly and slowly and thus are significantly more challenging.
> Deformable-3DGS and 4DGS fail on these casually captured monocular videos as demonstrated in our experiments including videos (from 0m12s to 0m56s) and tables (Tab.1, Tab.6, and Tab.7).
> - **Our contributions**. To address this challenge, MoDGS contains two novel techniques, i.e. 3D-aware-init and ordinal depth loss.
> 1) Gaussian splatting can be initialized from SfM points in static scenes but how to robustly initialize dynamic Gaussian fields is not well-studied.
> We show that the previous initialization method adopted in dynamic 3DGS does not work well in the real monocular setting in our experiments (Fig.7 and Tab.2).
> Thus, we propose 3D-aware-init which provides a robust starting point for the deformation field and canonical space Gaussians.
> 2) In the real monocular setting, we have to rely on monocular depth estimation due to the weak multiview constraints in casually captured videos. However, how to supervise the dynamic Gaussian fields with inaccurate monocular depth maps is still an open question.
> The proposed ordinal depth loss is designed to effectively utilize the inconsistent depth for the 4D reconstruction.
>
>
> ### W2: The experimental comparisons lack fairness. In most quantitative comparisons, this work is only compared against methods that require multi-view camera input. And only found comparative results of RoDynRF presented.
> ### A2:
> - **Choice of baseline methods.** We have tried our best to include all available competitive baseline methods including real monocular video methods, i.e. RoDynRF and Shape-of-Motion (arxvi:2407.13764, details in Appendix A.6), and the teleported monocular video methods, i.e. Dynamic-GS and SC-GS. We have demonstrated improved rendering quality than all these baseline methods.
>
> - **Why not report the quantitative results of RoDynRF?**
> Since the casually captured monocular videos only contain weak multiview constraints, all methods may have a different scene scale from the ground truth. We have tried to align the scale of RoDynRF with the ground truth but the alignment still fails in some cases. Thus, we provide the qualitative comparison (starting at 4m22s in the supplementary video), which shows that our method achieves much better rendering quality.

---

> > ### Author Response · Authors · 2024-11-28
> > **We are waiting for your further feedback. Let's discuss**
> >
> > Dear Reviewer **FCsY**,
> >
> > We appreciate your dedicated time and effort in reviewing our submission to ICLR. We have diligently worked to address your feedback in a thorough manner. If you have any further questions or require additional clarification, please do not hesitate to reach out to us. We are willing to provide further information and engage in a constructive discussion.
> >
> > Best regards,
> >
> > The authors

---

> ### Author Response · Authors · 2024-11-25
>
> We would like to express our sincere gratitude for dedicating your time and effort to reviewing our manuscript.  We have carefully considered and responded to all the concerns you raised in your review, as detailed in our response and the revised manuscript.
>
> As the Reviewer-Author discussion phase approaches its conclusion, we kindly await any further feedback you may have. Should you have any additional questions or require further clarification, we would be more than happy to provide detailed responses.
>
> Thank you once again for your valuable assistance.

---

> ### Author Response · Authors · 2024-12-02
>
> Dear Reviewer **FCsY**,
>
> Thanks for your valuable time reviewing our paper. The discussion session is closing soon. We are eager to hear your additional feedback of our paper. Thank you!
>
> Best Regards,
>
> The Authors

---

### Official Review · Reviewer_xth1 · 2024-11-03

**Soundness:** 4
**Presentation:** 3
**Contribution:** 4
**Rating:** 8
**Confidence:** 3

**Summary:**

The paper proposes MoDGS, a novel pipeline to render high quality novel views of dynamic scenes from casually captured monocular videos. Unlike traditional dynamic scene reconstruction methods that rely on rapid camera motions to establish multiview consistency, MoDGS is designed for videos with static/slowly moving cameras, where such consistency is weaker. The core of their method involves using a single-view depth estimation technique to guide scene learning and introducing a 3D-aware initialization method to construct a realistic deformation field. MoDGS incorporates an innovative ordinal depth loss to address the challenge of depth inconsistency across frames, enhancing the coherence and quality of rendered views. Experiments on datasets such as DyNeRF, Nvidia, and a self-collected dataset demonstrate it's ability to outperform SOTA methods in novel view synthesis, achieving superior image quality even in challenging dynamic scenarios.

**Strengths:**

MoDGS represents an original approach within  novel view synthesis  and dynamic scene modeling by specifically addressing the limitations of existing methods for casually captured monocular videos. The authors introduce a 3D-aware initialization mechanism and an ordinal depth loss, that offer a solution that successfully reduces the dependency on rapid camera motion. The novel use of ordinal depth loss to maintain depth order among frames, rather than relying solely on absolute values, represents an innovative perspective on addressing depth consistency issues, which has practical implications for improving depth coherence in dynamic scenes captured casually. I believe the paper is well-executed in terms of technical rigor, with comprehensive evaluations across three datasets: DyNeRF, Nvidia, and a newly created monocular casual video dataset. Each component of MoDGS is thoroughly tested and ablated to demonstrate its impact on the final results. This systematic experimentation supports the author’s claim that MoDGS significantly outperforms other approaches in the quality of novel-view rendering for dynamic scenes. The paper is structured logically, with clear explanations of each component of the MoDGS pipeline. The figures visually support the textual explanations, making complex concepts more understandable to a reader. The method  has significant implications for real-world applications that involve casually captured videos, such as mobile AR/VR, video editing, and 3D content creation. By enabling high-quality novel view synthesis from single-camera footage without multiview camera motions, MoDGS broadens the scope of dynamic scene reconstruction, making it accessible to a wider range of use cases. The method’s ability to handle both static and dynamic elements in monocular videos opens new avenues for monocular depth estimation and dynamic scene modeling, where single-camera approaches have been historically constrained by depth inconsistency issues.

**Weaknesses:**

While the ordinal depth loss is a novel way to improve depth coherence, I believe the paper may benefit from more discussion on its limitations. Specifically, the ordinal depth loss assumes a consistent depth order among frames, which may not hold in scenes with complex occlusions or reflections. MoDGS assumes smooth transitions between frames for consistent depth ordering. However, the approach may face challenges in scenes with rapid or erratic movement where objects appear and disappear frequently. While it performs well on scenes with relatively smooth dynamics, addressing how the method might be adapted or optimized for highly dynamic environments would improve its versatility. The method relies heavily on single view depth estimators to guide the reconstruction process. Although the depth estimation technique used is SOTA, it still inherits the limitations of single view estimators, particularly in complex scenes with specular surfaces or low-lit conditions. Including a more detailed analysis on how the quality of the depth estimator impacts the proposed method’s performance, and potentially exploring integration with other depth supervision methods could potentially make the approach more adaptable across varying input qualities.

**Questions:**

1. Can you elaborate on the choice of ordinal depth loss over other depth loss functions, such as perceptual depth consistency? How did the ordinal depth loss compare to other depth loss formulations in preliminary experiments, and what were the observed advantages or disadvantages?
2. How robust is MoDGS in scenarios with heavy occlusions or specular reflections? Would integrating additional priors or multi-scale depth estimations help in such cases?
3.   How does MoDGS compare with recent depth consistency techniques, particularly those used in self-supervised monocular depth estimation? Exploring this comparison could shed light on the effectiveness of the ordinal depth loss relative to existing methods.

**Details Of Ethics Concerns:**

A version of this paper is available on arxiv https://arxiv.org/pdf/2406.00434, and I had viewed a tweet earlier in the summer with the same title, paper, code: https://x.com/zhenjun_zhao/status/1798281777242632700. This may violate the double-blind review that is required, so I would like that to be known.

---

> ### Author Response · Authors · 2024-11-21
>
> We thank the reviewer for the positive and detailed review as well as the suggestions for improvement. Our response to the reviewer’s comments is below:
>
> ## Weakness
> ### W1. The paper may benefit from more discussion on its limitations. 1. The assumption of consistent depth orders among frames may not hold in scenes with complex occlusions or reflections. 2. MoDGS may face challenges in scenes with rapid or erratic movement. 3. The method depends on single-view depth estimators, inheriting their limitations in complex with specular surfaces or low-lit environments. A detailed analysis of estimator quality and integration with other methods could enhance adaptability.
>
> ### A1:
> - **Assumption of consistent depth orders.**
> We agree with reviewers that MoDGS assumes overall depth order consistency that can be satisfied by most depth estimators, GeoWizard, DepthAnything, etc. For these depth estimation methods, some flickering occurs in certain regions (as shown in our supplementary demo video), but the overall depth order remains consistent and thus our method successfully reconstructs dynamic Gaussian fields. Without this depth order consistency, our method could fail. We have added this discussion in the revision(L525).
>
>
> - **Challenges in rapidly changing scenes.**
>
> Rapidly moving scenes are indeed challenging for dynamic Gaussian field reconstruction. Such rapid motions may bring an inaccurate camera pose estimation, which is still challenging for MoDGS and all existing dynamic Gaussian reconstruction methods. We may combine the deblur methods with some motion prior to solving this in future works. Following your suggestions, we have added this limitation to the revised version(L527).
>
> - **The method depends on single-view depth estimators, inheriting their limitations in complex with specular surfaces or low-lit environments.**
>
> We agree with the reviewer that specular surfaces and low-lit conditions pose challenges for monocular depth estimation. In our experiments, we demonstrate a certain level of robustness in handling specular and low-lit scenes. For instance, MoDGS successfully reconstructs the windows in the Cook Spinach scene (from the DyNeRF dataset, with the video starting at 2m19s, which is also a low-lit scene), the windshield in the Truck scene (from the NVIDIA dataset, video starting at 2m54s), and the balloon in the Balloon1-2 scene (from Nvidia dataset, video start at 2m43s). However, for severely low-lighting or specular scenes, our method could fail due to the absence of reasonable depth estimation and we have added this in the limitation(L528).

---

> ### Author Response · Authors · 2024-11-21
>
> ## Question
> ### Q1: How about Comparing ordinal depth loss with other depth loss functions, such as perceptual depth consistency? what were the observed advantages or disadvantages?
> ### A1:
> Thanks for your suggestion. We present the results of this ablation study with perceptual (LPIPS) depth loss in Appendix A.16.
> Such LPIPS loss produces much better results than the vanilla Pearson loss, which demonstrates that such LPIPS loss is also insensitive to the absolute difference between depth values in some sense.
> Meanwhile, compared to the depth maps generated using the LPIPS loss, our maps are noticeably smoother and exhibit less noise.
> The reason may be that LPIPS loss is mainly trained on RGB images and performs less discriminative on the images converted from depth maps.
>
>
>
> ### Q2: How robust is MoDGS in scenarios with heavy occlusions or specular reflections?  Would integrating additional priors or multi-scale depth estimations help in such cases?
>
> ### A2:
> - **How to handle scenarios with heavy occlusions.** We agree that MoDGS may fail to handle scenarios with heavy occlusions. If the occluded regions are observed at some timestamps, MoDGS is able to accumulate information among different timestamps to complete these occluded regions on some timesteps as shown in Appendix A.4. However, MoDGS cannot generate new contents for the occluded regions, which indeed degenerate the rendering quality on occluded regions. Incorporating some generative priors such as diffusion models could alleviate this problem.
>
> - **Possible solutions dealing with specular scenarios.**
> MoDGS can reconstruct some specular objects but show some artifacts because MoDGS does not involve any special design for specular objects. We have added this discussion in the revision(L528). As shown by recent work [4], a potential solution for this is to introduce inverse rendering in dynamic scenes, which enables accurate modeling of specular surfaces in dynamic scenes.
>
>
>
>
> ### Q3: How about comparing with recent depth consistency techniques, particularly those used in self-supervised monocular depth estimation?
> ### A3:
> - Thank you for your suggestion. Our method is mainly targeted for the novel-view synthesis while we agree that MoDGS produces consistent video depth maps as a side product. Recent methods like [1,2,3] produce consistent depth maps but are targeted for depth estimation. We will conduct an experiment to compare the depth map quality of MoDGS with these baseline methods in the revision and we are still working on it. We will update the results once finished.
>
> [1] Luo et al.,  Consistent video depth estimation, ToG, 2020.
>
> [2] Wang et al.,  Neural Video Depth Stabilizer, ICCV, 2023.
>
> [3] Xu et al., Depthsplat: Connecting gaussian splatting and depth, arXiv preprint, 2024.
>
> [4] Fan et al., SpectroMotion: Dynamic 3D Reconstruction of Specular Scenes, arXiv preprint, 2024.

---

> ### Author Response · Authors · 2024-11-25
>
> Hi! Regarding Q3, we have finished the comparison with a recent work the neural video depth stabilizer (NVDS) [1]. The results are added in manuscripts (A.15, Fig 16) and the video in supplementary files (from  4m41s to 4m52s).  It can be observed that both depth maps are temporally consistent, but the depth maps of MoDGS contain more details than NVDS.
>
> - [1] wang et al., Neural Video Depth Stabilizer, ICCV, 2023.

---

> ### Author Response · Authors · 2024-11-25
>
> We would like to express our sincere gratitude for dedicating your time and effort to reviewing our manuscript.  We have carefully considered and responded to all the concerns you raised in your review, as detailed in our response and the revised manuscript.
>
> As the Reviewer-Author discussion phase approaches its conclusion, we kindly await any further feedback you may have. Should you have any additional questions or require further clarification, we would be more than happy to provide detailed responses.
>
> Thank you once again for your valuable assistance.

---

### Official Review · Reviewer_1rXU · 2024-11-03

**Soundness:** 4
**Presentation:** 4
**Contribution:** 4
**Rating:** 8
**Confidence:** 5

**Summary:**

The paper introduces MoDGS (Monocular Dynamic Gaussian Splatting), a novel approach for rendering dynamic 3D scenes from casually captured monocular videos, overcoming limitations faced by prior dynamic NeRF and Gaussian Splatting methods via depth estimation. These existing approaches require either extensive camera movement or synchronized multi-view setups to establish multiview consistency, which is lacking in casual, minimally moving videos.

To tackle this challenge, MoDGS incorporates recent advancements in single-view depth estimation to guide the learning of a deformation field that represents scene dynamics. The method introduces a novel ordinal depth loss to address the depth inconsistency in single-view depth maps, enhancing the robustness and continuity of 3D scene reconstruction.

Comprehensive experiments across multiple datasets (Nvidia, DyNeRF, DAVIS, and a self-collected casual video dataset) demonstrate that MoDGS produces high-quality novel views in dynamic scenes, outperforming state-of-the-art methods. The authors also plan to release their code and dataset to support future research in this area.

**Strengths:**

For the novelty, this paper makes a distinct contribution to introducing depth supervision into the domain of dynamic Gaussian Splatting (DGS) for monocular dynamic input. This approach is novel yet intuitive, filling a key gap in the field for cases where the input consists of casually captured videos with minimal camera movement. Compared to the other papers in the field that mechanically put all fancy complicated input feature streams or loss functions together, the proposed solution is conceptually straightforward but impactful, pushing forward the capabilities of monocular dynamic scene reconstruction.

The experiments are well-designed and executed, rigorously testing the proposed method across various datasets, including Nvidia, DyNeRF, and DAVIS. Each experiment logically supports the methodology, demonstrating how the 3D-aware initialization and ordinal depth loss contribute to enhanced depth consistency and scene deformation modeling. The results clearly show MoDGS’s robustness and superiority over baseline methods, adding confidence in its effectiveness.

The paper is presented with clarity and precision, making even technical aspects of the method easy to follow. The figures and tables are well-constructed and informative, providing visual clarity to support the text and helping to reinforce the main findings. The logical flow, from problem statement to method explanation and results, enables readers to understand the method's motivation and benefits seamlessly.

**Weaknesses:**

MoDGS is validated across several datasets, which demonstrates its robustness. However, the paper could discuss the potential limitations in generalizing this approach to different depth estimation models. It would demonstrate the robustness of the proposed method and its generalizability.

**Questions:**

From the perspective of a peer, I suggest the authors address the concept of 'Depth supervised in dynamic GS' in the title. After all, a novel method would be more informative and important to the other researchers than a usage scenario like ''casually captured video".

---

> ### Author Response · Authors · 2024-11-21
>
> We thank the reviewer for the positive feedback and constructive suggestions. Our response to the reviewer’s concerns is below:
>
> ## Weakness
> ###  W1:Potential limitations in generalizing this approach to different depth estimation models? MoDGS is validated across several datasets, which demonstrates its robustness. However, the paper could discuss the potential limitations in generalizing this approach to different depth estimation models. It would demonstrate the robustness of the proposed method and its generalizability.
>
> ### A1:
>
> We agree that the proposed method may fail when the prior depth estimation methods completely fail because casually captured monocular videos contain fewer multiview constraints and thus we rely on the depth prior to constrain geometry. However, current monocular depth estimators, like DepthAnythingv2, Depth Pro, GeoWizard, and Marigold, are powerful and can produce reasonable depth maps in most cases.
> We have added this discussion to the paper(L525) according to your suggestion.
>
> To further prove the robustness to different depth quality, in Appendix A.8, we present an ablation study on robustness to depth quality by introducing Gaussian noise into the input depth maps. The results demonstrate that our method is robust to a certain range of noise and is not sensitive to variations in depth quality. In Appendix A.11, we present an ablation study on the robustness to different depth estimation methods, it can be observed that with more stable video inputs, our ordinal depth loss still performs good results, which is better than the Pearson depth loss.
>
>
> ## Question
>
> ###  Q1:Suggestion about changing to a more informative title.
> ### A1:
> Thank you for your suggestion. We have changed it to "MoDGS: Dynamic Gaussian Splatting from Casually-captured Monocular Videos with Depth Priors".

---

> ### Author Response · Authors · 2024-11-25
>
> We would like to express our sincere gratitude for dedicating your time and effort to reviewing our manuscript.  We have carefully considered and responded to all the concerns you raised in your review, as detailed in our response and the revised manuscript.
>
> As the Reviewer-Author discussion phase approaches its conclusion, we kindly await any further feedback you may have. Should you have any additional questions or require further clarification, we would be more than happy to provide detailed responses.
>
> Thank you once again for your valuable assistance.

---

> ### Author Response · Authors · 2024-11-26
>
> Hi! Regarding W1, We conducted ablation studies using three additional depth estimation methods[1,2,3] and compared the performance of our MoDGS. The results are presented in Appendix A.20.  When we switched to different depth estimation methods, the PSNR varied by less than 0.35, the SSIM by less than 0.01, and the LPIPS by less than 0.015. These minimal changes demonstrate the robustness of our approach to varying distributions of real-world depth estimators. The main reason is that our method only relies on the order of depth values and most depth estimators are able to estimate correct depth orders in spite of their varying error types in different scenes. However, we also agree that the strong varying environments or cluttered scenes may break the correctness of predicted depth orders so our method may fail in these extremely challenging cases.
>
> Some qualitative visualization of depth inputs from different methods and depth maps rendered from MoDGS are added in manuscripts (A.20, Fig 17),  and supplementary video (from  4m52s to 5m08s).  It can be observed that MoDGS produces consistent video depth from different types of estimated depth maps.
>
>
> - [1] Ke et al., Repurposing diffusion-based image generators for monocular depth estimation, CVPR, 2024.
> - [2] Yang et al., Depthanything: Unleashing the power of large-scale unlabeled data, CVPR, 2024.
> - [3] Shao et al., Learning Temporally Consistent Video Depth from Video Diffusion Priors, arXiv preprint, 2024.

---

### Author Response · Authors · 2024-11-21

### summary:

- **Acknowledgement.**
We sincerely appreciate your valuable feedback on our work. We are delighted that you
that most of the reviewers have recognized our work’s strengths, including acknowledging the novelty and effectiveness of our 3D-Aware-init and ordinal depth loss (Reviewer 1rXU, xth1, Gzks), well-designed experiments(Reviewer 1rXU,xth1),  good experimental results (Reviewer 1rXU,xth1, FCsY), clear presentation(Reviewer 1rXU, FCsY). Thank you once again for your valuable feedback. In each response, we will thoroughly address your concerns, providing detailed explanations to clarify the points and answer your questions.



- **our revisions.**
To conclude: We added **five** new sections in the Appendix and added **four** limitations in Sec. 4.4.
1) In the A.15 section, we discuss our MoDGS and other depth consistency methods and analyze them from a knowledge distillation perspective.
2) In A.16, we present ablation studies about using perceptual loss to the rendered depth maps.
3) In A.17, We provide results on scenes with relatively complex motions to demonstrate the robustness of MoDGS.
4) In A.18, We show results on scenes with relatively rapid motions to explore the potential of applying MoDGS in such challenging settings.
5) In A.19, We discuss how to handle scenes with heavy occlusions and specular objects.
methods completely fail.
6) In the limitation section, we outline potential limitations concerning extremely low-light conditions, rapidly moving objects, specular scenes, and our depth order assumption.

---

> ### Author Response · Authors · 2024-11-25
>
> # **Our revisions summary:**
> Regarding our manuscripts, We newly added one new section in the Appendix and updated experiments results in A.15.   Regarding our demo video in supplementary files, we added two sections.
> ## Manuscripts:
> - In A.15,  we visually compare rendered depth maps using our MoDGS with stabilized depth maps from a recent video depth stabilization method.
> - In A.20, We present quantitative and qualitative results using different depth estimation methods, demonstrating our method's robustness.
>
> ## Demo video in supplementary:
> - From 4m42s to 4m52s, we present visual comparison of rendered depth maps with a video depth stabilization method.
> - From 4m52s to 5m08s, we showcase a visual comparison using four different depth estimation methods.

---

### Meta-Review · Area_Chair_unMd · 2024-12-15

**Metareview:**

This paper presents MoDGS, a dynamic scene modeling algorithm using casually captured monocular videos. To overcome the rapid camera motion assumption in previous methods, it integrates single-view depth estimation, 3D-aware initialization, and robust ordinal depth loss, and it shows superior scene reconstruction and rendering performance.

The reviewers found the proposed components to be technically novel and interesting. The 3D-aware initialization improves the consistency of the initial Gaussians, and the ordinal depth loss can handle the problem of depth consistency across frames. The ablation study and experimental validations are provided at a sufficient level, and the paper is well written and clearly structured.

The reviewers also found that the proposed method builds incrementally on existing approaches such as deformable 3DGS by optimizing depth consistency and deformation, and its performance depends significantly on external single-view depth estimators. The influence of these models is not fully analyzed, especially in challenging conditions such as low illumination or complex scenes, and this may be the main limiting factor of the proposed algorithm, especially in complex scenes where monocular depth estimation becomes unreliable. Also, the comparison with multi-view methods in a monocular video setting may not be a fair comparison.

During the rebuttal period, the authors provided answers to the reviewers' questions and performed additional experiments with different depth estimators.
All reviewers agree that the technical contribution of the proposed algorithm is significant and the paper is clearly written, and most reviewers agree to accept the paper.

**Additional Comments On Reviewer Discussion:**

Since most reviewers were positive about the paper and the only negative reviewer FCsY did not participate in the discussion. The authors provided the explanation on the issues raised by FCsY during the rebuttal and the AC finds it reasonable.

---

### Decision · Program_Chairs · 2025-01-22

Accept (Poster)